# How cells determine the number of polarity sites

**Jian-geng Chiou, Kyle D Moran, Daniel J Lew\***

Department of Pharmacology and Cancer Biology, Duke University Medical Center, Durham, United States

**Abstract** The diversity of cell morphologies arises, in part, through regulation of cell polarity by Rho-family GTPases. A poorly understood but fundamental question concerns the regulatory mechanisms by which different cells generate different numbers of polarity sites. Mass-conserved activator-substrate (MCAS) models that describe polarity circuits develop multiple initial polarity sites, but then those sites engage in competition, leaving a single winner. Theoretical analyses predicted that competition would slow dramatically as GTPase concentrations at different polarity sites increase toward a 'saturation point', allowing polarity sites to coexist. Here, we test this prediction using budding yeast cells, and confirm that increasing the amount of key polarity proteins results in multiple polarity sites and simultaneous budding. Further, we elucidate a novel design principle whereby cells can switch from competition to equalization among polarity sites. These findings provide insight into how cells with diverse morphologies may determine the number of polarity sites.

**\*For correspondence:**
daniel.lew@duke.edu

**Competing interests:** The authors declare that no competing interests exist.

## Introduction

Eukaryotic cells display a very wide diversity of cell morphologies, which are often critical to carry out specialized cell functions. Different morphologies arise through specific arrangements and actions of the cytoskeleton. In turn, the cytoskeleton is regulated by the conserved Rho family of GTPases (*Etienne-Manneville and Hall, 2002*). A subset of these GTPases (Cdc42, Rac, Rop) regulates cell polarity by concentrating at one or more regions of the plasma membrane (*Park and Bi, 2007*; *Wu and Lew, 2013*). In many cell types (e.g. migrating cells, plant pollen tubes, and root hairs or diverse budding yeasts including *Saccharomyces cerevisiae*, *Candida albicans*, *Cryptococcus neoformans*, and *Ustilago maydis*), it is crucial to maintain one and only one polarity domain, establishing a single polarity axis (front) that leads to movement or growth in that direction (*Chiou et al., 2017*; *Houk et al., 2012*; *Wu and Lew, 2013*; *Yang and Lavagi, 2012*). In other cell types (e.g. neurons with many neurite tips, plant cells that form xylem, or filamentous fungal cells with branches), multiple active-GTPase polarity sites coexist in the same cell (*Dotti et al., 1988*; *Knechtle et al., 2003*; *Oda and Fukuda, 2012*). These differences raise the question of how Rho-GTPase polarity systems in specific cell types can be tuned to yield the desired number of polarized fronts.

It has long been appreciated that interacting biochemical networks that include positive feedback and differential diffusion of reactants can generate systems capable of spontaneous pattern formation (*Meinhardt, 2008*; *Meinhardt and Gierer, 1974*; *Turing, 1952*). Key properties of pattern formation by polarity GTPase systems can be captured by a subset of such systems that we refer to as Mass Conserved Activator Substrate (MCAS) models (*Brauns et al., 2020b*; *Chiou et al., 2018*; *Goryachev and Pokhilko, 2008*; *Halatek et al., 2018*; *Jilkine and Edelstein-Keshet, 2011*; *Mori et al., 2008*; *Otsuji et al., 2007*; *Otsuji et al., 2010*). These models consist of sets of partial differential equations (PDEs) that encode the interconversion of polarity factors between two forms: a membrane-bound (and hence slow-diffusing) 'activator' and a cytosolic (and hence rapidly diffusing) 'substrate' (*Figure 1A*). One critical feature of these systems is positive feedback,

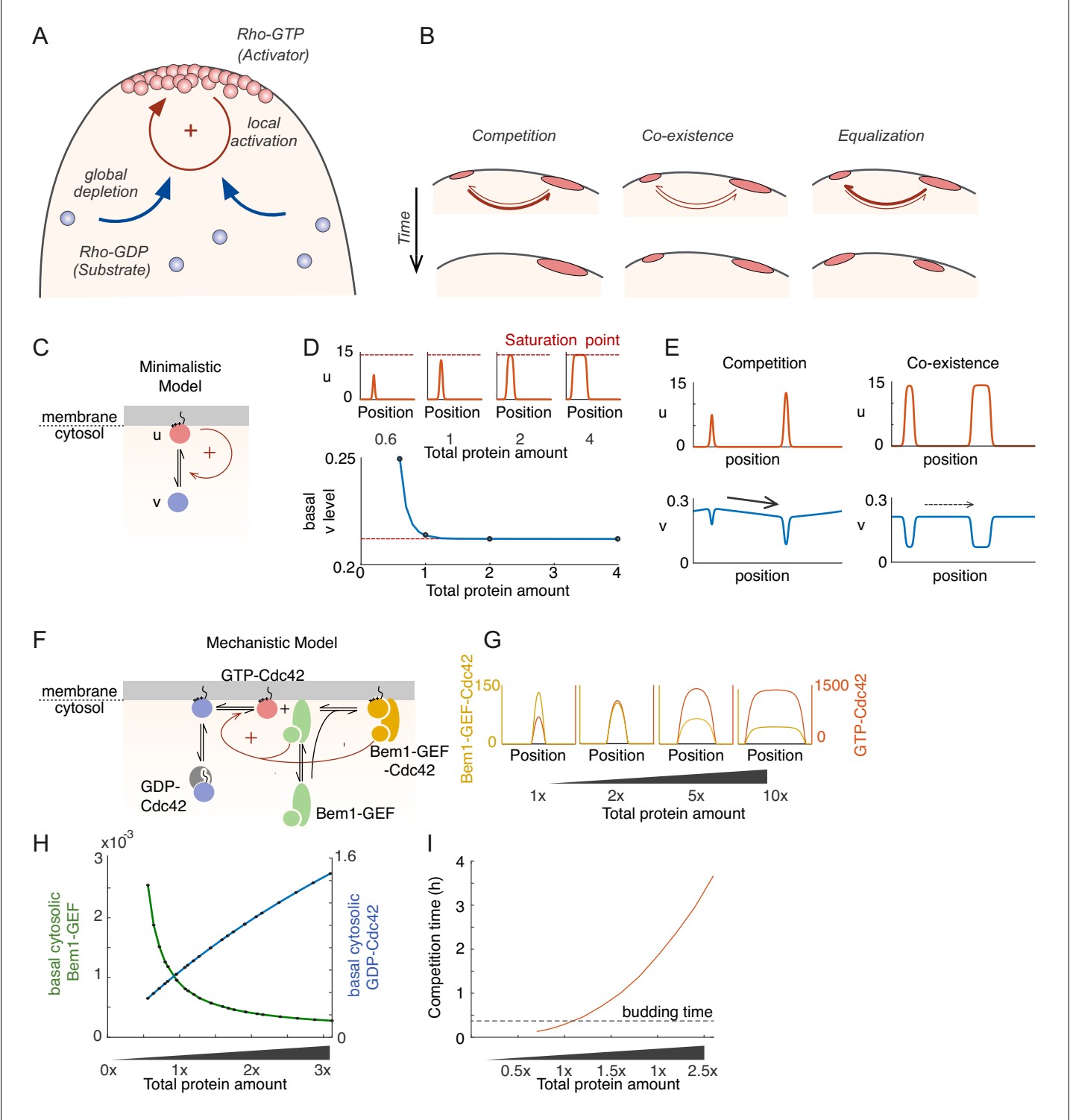

**Figure 1.** Competition in a mechanistic Rho-GTPase model. (**A**) Rho-GTPase polarity circuits can be modeled as mass-conserved reaction-diffusion systems where membrane-bound Rho-GTP (red) is a slow-diffusing activator and cytosolic Rho-GDP (blue) is a fast-diffusing substrate. Such systems polarize Rho-GTPase to spatially confined polarity sites based on positive feedback (+) via local activation that recruits substrate from the cytoplasm, leading to global substrate depletion. (**B**) A starting condition with two unequal peaks of activator can evolve in three ways. Competition occurs if the larger peak recruits substrate better than the smaller; coexistence occurs if both peaks recruit substrate equally well; and equalization occurs if the smaller peak recruits substrates better than the larger. (**C**) Schematic of the minimalistic MCAS model with one Rho-GTPase converting between activator (u: Rho-GTP, red) and substrate (v: Rho-GDP, blue) forms. Arrows depict reactions. The positive feedback is highlighted in red (see text for

*Figure 1 continued on next page*

*Figure 1 continued*

details). (**D**) In the minimalistic model, as total protein amount in the system increases, peak u concentration (red) approaches a saturation point while basal v concentration (blue) declines to a limit. Basal here refers to the concentration at the edge of the peak. 1x protein amount equivalent to starting uniform v concentration of 2. (**E**) Snapshots of simulations starting from two single peak steady states placed next to each other in the same domain. With peaks far from saturation, the larger peak depletes more substrate than the smaller (Left panel, starting protein amounts 0.6x, 1x), leading to a net flux of substrate (arrow) that results in competition. With peaks close to saturation, substrate depletion is similar for both (Right panel, starting protein amounts 2x, 4x), leading to little flux and therefore coexistence. (**F**) Schematic of a mechanistic model of the yeast polarity circuit: arrows depict reactions assumed to occur with mass-action kinetics. The positive feedback is highlighted in red. (**G**) Increasing total protein amount saturates the peaks similarly to D. (**H**) Basal cytosolic level of the limiting species Bem1-GEF declines to a limit while that of Cdc42 increases with increased total protein amount. (**I**) Competition time increases as peaks saturate. Competition was started from two insulated 1-peak steady states each containing 60% and 40% of the total proteins. Competition time is defined by the time it takes to evolve from 70%:30% (relative Cdc42T amounts between two peaks) to 99%:1%.

The online version of this article includes the following figure supplement(s) for figure 1:

**Figure supplement 1.** Effects of increasing the abundance of individual polarity factors (Cdc42 or Bem1-GEF) in mechanistic polarity model.

such that membrane regions with higher concentrations of activator can locally recruit and activate more substrate from the cytoplasm. A second critical feature is the difference in diffusivity between the activator and the substrate, allowing a localized accumulation of activator to recruit substrate from a much wider region of cytoplasm. A third critical feature is mass conservation: because the combined amount of activator and substrate is fixed, accumulation of activator at the membrane depletes substrate from the cytoplasm, limiting the size, activator concentration, and potentially the number of permissible activator-enriched regions.

MCAS models at the homogeneous steady state can develop inhomogeneous activator distributions either spontaneously through Turing instability, or in response to external cues. Once a local region becomes enriched for activator, it grows (acquires more activator) by recruiting more activator from the cytoplasm, eventually depleting cytoplasmic substrate levels until the system reaches a polarized steady state with a local peak in activator concentration. However, the fate of any given peak depends on the presence of other peaks, which can also deplete substrate from the cytoplasm. Peaks that differ in amount of activator would also differ in their ability to recruit cytoplasmic substrate (*Chiou et al., 2018*). A hypothetical case with two initial unequal peaks could evolve in three possible directions (*Figure 1B*): (1) Competition: If the peak with more activator grows more rapidly, then as cytoplasmic substrate levels become depleted, the smaller peak would be starved of the fuel it needs to survive, and begin to shrink, losing activator until there is only one peak at steady state. (2) Coexistence: If the two unequal peaks both grow at the same rate, they would persist indefinitely. (3) Equalization: If the peak with less activator grows more rapidly, then that would continue until the two are equal.

To capture essential behaviors of MCAS systems, previous research conducted in-depth mathematical analyses on minimalistic one-activator, one-substrate MCAS models. These analyses indicated that the mass conservation feature enforces the competition scenario, with the largest peak eventually becoming the only one (*Brauns et al., 2020b*; *Chiou et al., 2018*; *Ishihara et al., 2007*; *Otsuji et al., 2007*; *Otsuji et al., 2010*). However, recent modeling studies have shown that the growth rate of a peak 'saturates' as the activator in the peak exceeds a threshold. If more than one peak saturates, then the system switches to a coexistence scenario for biologically relevant timescales. This conclusion is general for minimalistic MCAS models, and the degree of saturation is the dominant factor determining uni- or multi-polar outcomes (*Brauns et al., 2020a*; *Chiou et al., 2018*). Equalization does not appear possible in minimalistic MCAS models, but has been observed in more complex models of some polarity circuits (*Howell et al., 2012*; *Jacobs et al., 2019*). The basis for equalization in such models is not well understood, and it has not yet been determined whether saturation or equalization occurs in cells.

One well-studied experimental system ideal for testing theoretical predictions of MCAS models is the budding yeast *Saccharomyces cerevisiae*. In yeast, the polarity GTPase Cdc42 cycles between a slow-diffusing GTP-bound form and a rapidly diffusing GDP-bound form. GTP-Cdc42 binds effector p21-activated kinases (PAKs)(*Bose et al., 2001*; *Cvrcková et al., 1995*; *Lamson et al., 2002*; *Zhao et al., 1995*), which bind the scaffold protein Bem1 (*Bose et al., 2001*; *Leeuw et al., 1995*), which binds the GEF Cdc24 that activates Cdc42 (*Bose et al., 2001*; *Butty et al., 2002*; *Ito et al.,*

*2001*; *Peterson et al., 1994*; *Rapali et al., 2017*). These interactions allow GTP-Cdc42 to recruit its own GEF, activating neighboring Cdc42 to yield positive feedback (*Johnson et al., 2011*; *Kozubowski et al., 2008*). This mechanism of positive feedback has recently been powerfully supported by findings of optogenetic approaches in several fungi (*Lamas et al., 2020*; *Silva et al., 2019*; *Witte et al., 2017*). Competition between polarity peaks has been observed experimentally in yeast, leading to development of a single Cdc42-enriched cortical region that generates a single bud in every cell cycle (*Howell et al., 2012*; *Witte et al., 2017*; *Wu et al., 2015*).

Here, we show that consistent with MCAS model predictions, larger yeast cells can generate multiple buds, in a manner that depends on the dosage of polarity genes. Furthermore, we identify a key feature in mechanistic MCAS models that enables equalization, and elucidate the underlying theoretical basis. By imaging polarization in cells that make multiple buds, we find that coexistence is the predominant mechanism that yields multi-budded yeast cells.

## Results

### A mechanistic model for the yeast Cdc42 system exhibits saturation, and switches from competition to coexistence as protein levels increase

In minimalistic models, positive feedback ensures that peaks with more activator are better at recruiting substrate than peaks with less, and therefore that larger peaks deplete cytoplasmic substrate more effectively than smaller peaks. Thus, when peaks of unequal size are present, the larger peak more effectively depletes the substrate, creating a cytoplasmic substrate gradient toward the larger peak that drives competition. However, as the amount of activator in a peak increases, the ability to recruit substrate eventually saturates, yielding a plateau in basal cytoplasmic substrate level (*Chiou et al., 2018*; *Figure 1D*). Note that at steady state, there is a local dip in cytoplasmic substrate concentration that mirrors the local peak in activator at the membrane: here we refer to the 'basal' substrate level as the substrate concentration that is reached outside this dip (i.e. the substrate concentration just outside the peak). It is a difference in the basal substrate levels between two peaks that can drive a flux of substrate from one peak to the other. But, as peaks approach saturation, the cytoplasmic substrate gradient becomes negligible, resulting in apparent coexistence (*Figure 1E*). This general result applies to both 1D and 2D models, although timescales of competition may differ (*Chiou et al., 2018*). However, it was unclear whether more complex MCAS models with multiple species would similarly exhibit saturation.

We first asked whether a multi-species mechanistic 2D model of the budding yeast system (*Goryachev and Pokhilko, 2008*; *Wu et al., 2015*) behaved in a similar manner to the minimalistic models in terms of saturation. The mechanistic model explicitly considers two species analogous to 'activators' at the membrane (GTP-Cdc42 and the Bem1-GEF-Cdc42 complex), either of which can be considered as promoting positive feedback through the other. The model also considers two cytoplasmic species as their respective 'substrates' (GDP-Cdc42 and Bem1-GEF), as well as other intermediate species with different characteristics (GDP-Cdc42 and Bem1-GEF at the membrane) (*Figure 1F*). When the total amounts of both Cdc42 and Bem1-GEF were increased in parallel, the cytoplasmic substrate Bem1-GEF was depleted and quickly reached a plateau (*Figure 1G,H*), similar to substrate depletion in the minimalistic model. In contrast, levels of the cytoplasmic substrate Cdc42 increased, indicating that this substrate was in excess and failed to saturate (*Figure 1H*). Increasing the amount of Cdc42 or Bem1-GEF complex individually also resulted in substrate depletion, but the species exhibiting saturation changed depending on which protein was added (*Figure 1—figure supplement 1*). We conclude that similar to minimalistic models, the mechanistic model exhibits saturation. However, in a multi-component system saturation is only seen for the limiting species. We note that the limiting species may not be the less abundant one, as other parameters of the model also influence which species becomes limiting.

We next asked whether saturation leads to coexistence in the mechanistic model. When simulations were initiated with two unequal peaks, competition occurred rapidly with low protein amounts, but slowed in a non-linear manner as the total amount of protein in the system was increased (*Figure 1I*), generating competition times that would be long compared to the yeast bud emergence timescale. In summary, saturation is also evident in a mechanistic model with multiple species, and

as in minimalistic models, the approach to saturation slows competition, driving the system toward coexistence.

## Testing model predictions: multi-polar growth in large yeast cells

A simple prediction emerging from the polarity models discussed above is that cells should switch from competition to coexistence as the amount of polarity 'activators' in the peaks increase. However, raising total protein concentrations in MCAS models generally drives them into a regime where Cdc42 activation occurs over the entire surface, yielding a uniform (depolarized) steady state. Indeed, previous work indicated that extreme overexpression or global optogenetic activation of polarity proteins leads primarily to depolarization, with only occasional multi-polarity (*Howell et al., 2012*; *Howell et al., 2009*; *Lamas et al., 2020*; *Silva et al., 2019*; *Witte et al., 2017*; *Ziman and Johnson, 1994*). A more robust way to generate multiple polarity domains would be to increase the size of cell while keeping overall protein concentrations constant (*Chiou et al., 2018*; *Ishihara et al., 2007*): this should lead to multi-budded cells while avoiding depolarized outcomes. One way to increase cell size is to arrest the cell cycle, but in yeast this approach leads to cytoplasm dilution as biosynthesis fails to keep pace with volume growth (*Neurohr et al., 2019*). Instead, we utilized cytokinesis-defective yeast mutants to obtain large connected cells that continue cycling but retain a normal overall protein composition.

The temperature-sensitive septin mutant *cdc12-6* is defective in cytokinesis at 37°C, generating chains of elongated, connected cells (*Figure 2A*, *Video 1*; *Hartwell, 1971*). Control experiments confirmed the continuity of the cytoplasm between the connected cells (*Figure 2—figure supplement 1A*), and showed that the concentration of polarity proteins remained constant as the cells grew larger (see below). In the first cell cycle after switching to 37°C, all cells generated a single bud, which remained connected to the mother. In the second cell cycle, most cells formed a single bud despite having two cell bodies and two nuclei. This is consistent with the idea that these larger cells retained an effective competition mechanism that yields only a single winning polarity site to produce the bud. However, a few cells generated two buds simultaneously. The fraction of multi-budded cells increased dramatically in the third and the fourth cell cycles (*Figure 2B*). Similar multi-budded outcomes were observed in a conditional cytokinesis-defective *iqg1* strain (*Figure 2—figure supplement 1B*; *Shannon and Li, 1999*), indicating that the phenotype is not specific to septin mutants. These results are consistent with the hypothesis that larger cells with more polarity proteins (due to their larger volume) can trigger a transition from competition to coexistence.

In MCAS models, the entire cell is diffusionally connected, allowing fluxes of polarity substrates in the cytoplasm. However, the geometry of septin-mutant cells raised the possibility that the undivided mother-bud neck might impose a diffusion barrier that suffices to yield multi-polarity. The narrowing at the neck is predicted to slow diffusional fluxes between cell bodies by about 20%, and this mild effect was indeed detected experimentally using photo-bleaching (*Figure 2—figure supplement 2*). To distinguish whether increased volume or slower diffusion from neck geometry is the dominant contributing factor for multipolar outcomes, we compared haploid and diploid mutant cells. Diploid cells have larger volume (and hence more polarity proteins) but also wider necks (which would provide a smaller impediment to diffusion) compared to haploids (*Figure 2C*). Despite having wider necks, the larger diploids generated significantly more two-budded cells than did the haploids (*Figure 2D*). This was due to cell size and not mating type, as MAT𝝰/MAT**a** cells (cells that have the large size of a diploid but the mating type of a haploid) behaved similarly to normal diploids (*Figure 2D*). Taken together, our findings indicate that the large cells generated following failure of cytokinesis can yield multi-budded outcomes, and that such outcomes can be predominantly attributed to the larger cell size.

## Competition in cytokinesis-defective cells can become ineffective

Larger cell size could lead to a multi-polar outcomes by affecting the number of polarity sites that initially form during polarity establishment, or the subsequent competition between sites, or both. To evaluate these features, we introduced the polarity probe Bem1-GFP (*Howell et al., 2012*), and employed a *cdc12-6 rsr1Δ* genetic background to avoid complexities associated with Rsr1-mediated bud-site-selection, which biases the location of polarity sites and slows competition by unknown mechanisms (*Bi and Park, 2012*; *Wu et al., 2013*). Control experiments indicated that *cdc12-6 rsr1Δ*

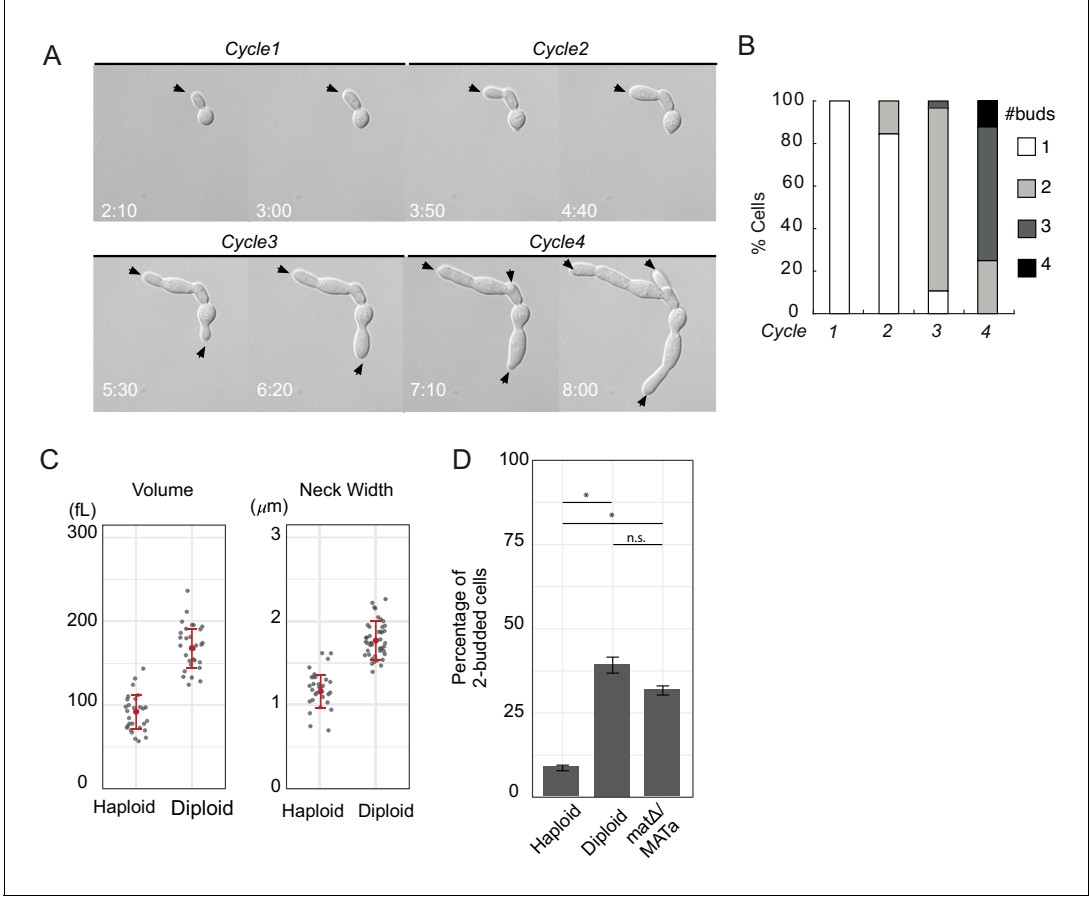

**Figure 2.** Large yeast cells can generate multiple buds simultaneously. (**A**) DIC time lapse movie of a haploid cytokinesis-defective *cdc12-6* mutant (DLY20240) over four budding cycles at restrictive temperature (37°C). Black arrows indicate growing buds. Time in hr:min. (**B**) The number of buds generated in each cell cycle was scored for N = 35 cells. (**C**) Volume and neck width of diploid (DLY20569) and haploid (DLY9455) *cdc12-6* cells in the second cell cycle at restrictive temperature. Red dot and intervals indicate mean and standard deviation. (**D**) Percentage of two-budded cells observed in the second cell cycle at restrictive temperature for diploid (DLY20569) and haploid (DLY9455) *cdc12-6* cells (Fisher's exact test $p<10^{-5}$). *matΔ*/MATa cells (DLY22887) are diploid but have a haploid mating type (Fisher's exact test p=0.0431 compared to diploid; $p<10^{-5}$ compared to haploid). Asterisks indicate $p<10^{-5}$ and n.s. indicates p>0.01. Error bars indicate SEM.

The online version of this article includes the following figure supplement(s) for figure 2:

**Figure supplement 1.** Multi-lobed *cdc12-6* and Iqg1-shutoff cells have continuous cytoplasms.

**Figure supplement 2.** The effect of the mother-bud neck on cytoplasmic diffusion.

mutants generated increasing numbers of multi-budded cells with increasing cell size, although fewer compared to the *cdc12-6 RSR1* cells (*Figure 3—figure supplement 1*). In addition to polarity sites, Bem1 could localize to attempted cytokinesis sites, but these were easily distinguishable by cell cycle timing (*Figure 3—figure supplement 2*).

As cells entered the second cell cycle after switching to 37°C, Bem1 became concentrated at one or two initial polarity sites, but cells with two initial sites often made only one bud, presumably as a result of competition (*Figure 3A*). Competition was evident both within individual cell compartments and between cell compartments connected by necks. However, in some cases, the initial sites persisted and gave rise to two buds. As cells entered the third cell cycle, Bem1 sometimes localized to three initial sites. The outcome in these cases was variable, with competition leaving one or two sites that gave rise to buds (*Figure 3B*). Thus, unipolar versus multipolar outcomes depend both on the number of initial polarity sites that form and on whether they are subsequently eliminated by competition.

## Increasing polarity protein abundance decreases the effectiveness of competition

To assess how cell size might affect the number of initial polarity sites, we compared *cdc12-6 rsr1Δ* cells in the second cell cycle versus the third cell cycle after switching to 37°C, as well as haploid and diploid mutant cells. As expected, third-cycle cells were larger than second-cycle cells, and diploids were larger than haploids (*Figure 4A*). The number of initial sites formed by each cell increased in a manner correlated with cell length (*Figure 4B*), suggesting that cell length can influence initial polarization. Variability in the number of initial sites that form is expected for a process that depends on inhomogeneities in the initial polarity protein distribution stemming from molecular noise to initiate polarization. However, the locations at which initial sites formed were non-random, with a preference for bud tips and mother cell locations (*Figure 4C*), suggesting that (even without Rsr1) polarity may not initiate purely from molecular noise, and that geometric factors may bias the process. When cells formed two initial sites, the distance between the sites was highly variable (*Figure 4D*). This is contrary to the expectation

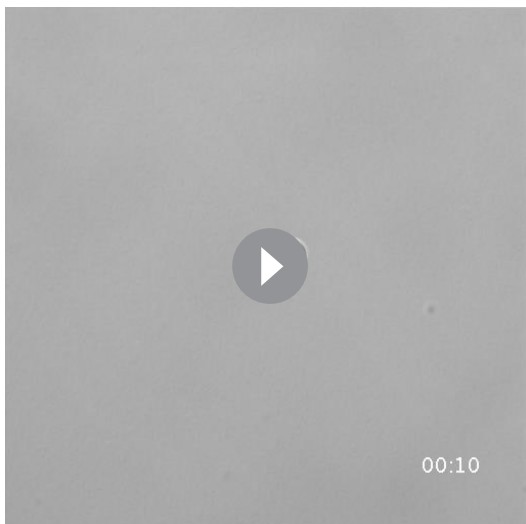

**Video 1.** Large yeast cells can generate multiple buds simultaneously. Time-lapse DIC images of a pair of *cdc12-6* cells (DLY20240) growing at restrictive temperature for four-cell cycles. The mother-daughter pair of cells were shifted to 37°C when the video begins. Time stamp indicates hr:min.

https://elifesciences.org/articles/58768#video1

for classical Turing-type models, which tend to form peaks separated by a characteristic length scale (*Meinhardt, 2008*). However, it is consistent with predictions from MCAS models that have no preferred inter-peak length scale (*Goryachev and Leda, 2020*; *Goryachev and Pokhilko, 2008*; *Halatek et al., 2018*).

To focus on the outcome of competition, we narrowed our analysis to the cells that established just two initial sites. Of these cells, the fraction that yielded two-budded outcomes increased with cell size (*Figure 4E*). Notably, similar-sized cells within the same population could differ their eventual outcomes (*Figure 4E*). Such variability may reflect the variability in protein concentrations between cells, or stochastic differences in the size of the initial peaks that formed (with similar peaks coexisting and unequal peaks competing). Overall, the probability of successful competition between polarity sites decreased as cells grew larger, consistent with a transition from competition to coexistence regimes.

In MCAS models, the main effect of larger size on system behavior is due to the increased total abundance of polarity proteins in the system, rather than the increased length or volume per se (*Chiou et al., 2018*). To ask whether this was also the case in yeast cells, we integrated additional copies of the *CDC42*, *CDC24* (GEF), and *BEM1* genes into our mutant cells. Doubling the gene copy number led to the expected increase in the concentration of the encoded proteins (*Figure 4—figure supplement 1*). An extra copy of any single gene did not greatly affect the frequency of multi-budded cells produced by *cdc12-6 rsr1Δ* mutants (*Figure 4—figure supplement 2*), consistent with previous overexpression reports in other strain backgrounds (*Freisinger et al., 2013*; *Howell et al., 2012*; *Howell et al., 2009*). We reasoned that by altering the stoichiometry of the polarity genes relative to each other, we might have changed the identity of the limiting species. To avoid that, we made a strain with double the gene copy number for each of the three genes. This did not affect the morphology, volume, or length of the cells (*Figure 4F*), and there was no systematic effect on the number of initial polarity sites formed (*Figure 4G*). However, among the cells that established two initial sites, the frequency of multi-polar outcomes was significantly increased (*Figure 4H*). We conclude that increasing the abundance of polarity proteins is sufficient to decrease the effectiveness of competition, yielding multi-polar outcomes.

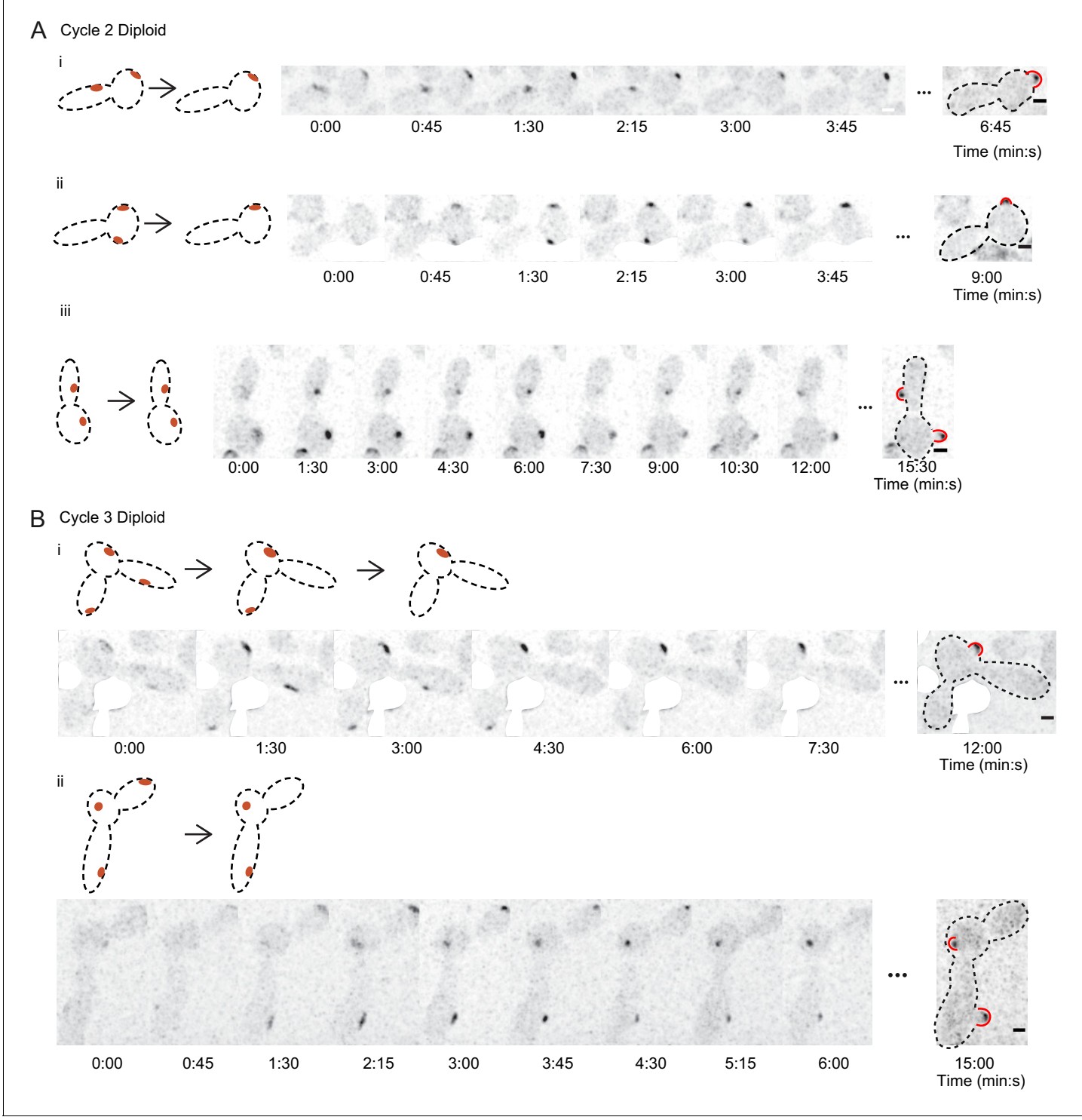

**Figure 3.** Polarity sites in large *cdc12-6* cells. (**A**) Cells that start with two polarity sites can show competition or coexistence. Example cells demonstrating typical Bem1 behaviors in *cdc12-6 rsr1* Δ diploids (DLY15376) in the second G1 phase after switching to restrictive temperature. Cartoons (left) depict the dynamics of polarity sites (red). Red bulges (right) indicate buds (note that buds can emerge in directions that are not within the focal plane). Cells i and ii: competition yields one bud. Cell iii: coexistence yields two buds. (**B**) Example cells from the third G1 phase after switching to restrictive temperature. These cells initially had three polarity sites but made only one (cell i) or two (cell ii) buds. Scale bar = 2 μm. White regions cover auto-fluorescent dead cells.

The online version of this article includes the following figure supplement(s) for figure 3:

**Figure supplement 1.** Large *rsr1Δ* cells can generate multiple buds similar to *RSR1* cells.

*Figure 3 continued on next page*

*Figure 3 continued*

**Figure supplement 2.** Timing of Bem1 localization in the cell cycle distinguishes abortive cytokinesis sites from polarity sites.

**Figure supplement 3.** Bem1 but not Cla4 localizes to abortive cytokinesis sites as well as polarity sites in *cdc12-6* mutants.

## Equalization is enabled by an indirect pathway from activator to substrate

The experimental results thus far are consistent with a switch from competition to coexistence, as predicted by the minimalistic MCAS model. However, multi-component MCAS models that incorporate negative feedback as well as positive feedback can yield a qualitatively different behavior, in which starting unequal peaks become equal. The first report to show equalization considered a mechanistic model of yeast polarity that incorporated a hypothesized negative feedback pathway (*Howell et al., 2012*). An intuitive explanation for such equalization is that larger peaks are penalized by generating more negative feedback, allowing smaller peaks to compete successfully (*Jacobs et al., 2019*). However, equalization behavior does not necessarily follow from the presence of negative feedback (*Chiou et al., 2018*; *Jacobs et al., 2019*), suggesting that this intuition is insufficient to account for equalization. Here, we sought to understand the features of polarity models that enable equalization.

Addition of a negative feedback loop to a minimalistic model (*Figure 5Ai*) did not enable equalization (*Chiou et al., 2018*). However, addition of a negative feedback via activation of a Cdc42 GAP (*Figure 5Aii*; *Jacobs et al., 2019*) did enable equalization. Addition of a negative feedback to our mechanistic model (*Figure 1F*) via inhibition of the GEF (simplified in *Figure 5Aiii*. Complete scheme in *Figure 5—figure supplement 1F*) also enabled equalization (*Figure 5B*). This mechanism was suggested by more recent experimental findings that Cdc42 promotes inhibitory phosphorylation of its GEF in yeast (*Kuo et al., 2014*). These findings suggested that some feature(s) absent from the first model (*Figure 5Ai*) but shared by the others (*Figure 5Aii–iii*) might explain equalization. One such feature is the addition of a new species (the GAP in model ii and the inhibited GEFi in model iii).

The GAP and the inhibited GEFi are neither substrates nor activators, and appear to play different roles in the polarity circuit. However, we noticed that they both provide a source of substrate: the GAP converts local GTP-Cdc42 into the substrate GDP-Cdc42, while the inhibited GEFi turns into the substrate GEF upon dephosphorylation. Thus, in both cases a new species produced by the activator is highly mobile and generates a substrate in the cytoplasm. We reasoned that a larger peak of activator would generate more of this new species (GAP or GEFi) in its vicinity, and by generating more substrate this new species might reverse the concentration gradient of substrate in the cytoplasm, driving a flux of substrate toward the smaller peak to yield equalization.

Interestingly, the key to equalization in the hypothesis proposed above is not negative feedback per se, but rather the existence of a new species created by an activator that can generate a substrate. To test our hypothesis, we modified the minimalistic model to include an 'indirect substrate' species (*Figure 5C*): In addition to direct conversion of the activator $u$ to the substrate $v$, $u$ can also be converted to the indirect substrate $v_i$, which can then be converted to $v$. We made the $u \rightarrow v_i$ reaction linear with $u$, such that this model lacks the non-linear negative feedback present in the models discussed above, allowing us to probe whether equalization arises due to the presence of indirect substrate even without such negative feedback.

The new indirect substrate model recapitulated the switch from competition to equalization behavior as the total protein amount in the system was increased. Previously, at the single peak steady states of the minimalistic model (*Figure 1D*), basal levels of the cytoplasmic substrate $v$ decreased until they reached a limit as the total protein amount in the system was increased. However, in the indirect substrate model, basal levels of $v_i$ rose steadily as the total protein amount was increased (*Figure 5D*). The basal level of substrate $v$ initially decreased, but then rose as total protein amount was increased (*Figure 5E*), presumably due to flux from $v_i$. When two unequal peaks were placed together, the system evolved either to a single peak (competition) or to two equal peaks (equalization), depending on the amount of protein in the system (*Figure 5F,G*). A key factor appeared to be the relative basal levels of cytoplasmic substrate associated with each peak: when the larger peak was associated with lower substrate levels, the system displayed competition; when the smaller peak was associated with lower substrate levels, the system displayed equalization.

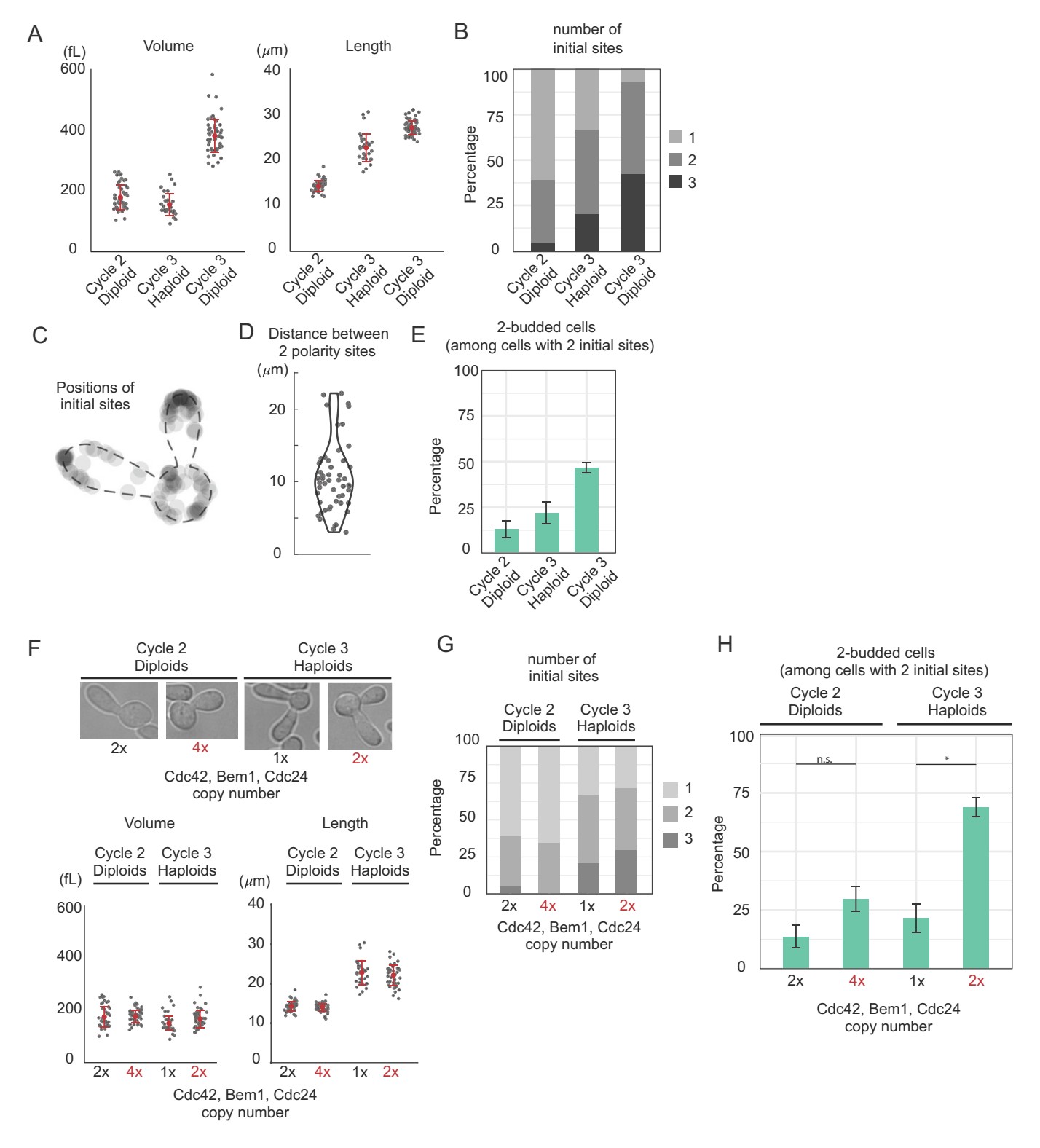

**Figure 4.** Increasing abundance of polarity proteins enhances the frequency of multipolar outcomes. (**A**) Cell volume and length in three populations of *cdc12-6 rsr1Δ* cells at restrictive temperature: diploids (DLY15376) in the second or third cell cycle and haploids (DLY9453) in the third cell cycle. Red dot and interval indicate mean and standard deviation. (**B**) The number of initial polarity sites in each population. (**C**) Initial sites form at non-random positions. N = 123 initial polarity sites in cycle three diploids were mapped onto a generic cell outline. (**D**) The distance between polarity sites is highly variable. (**E**) Percentage of two-budded cells (bipolar outcomes) among those that established two initial sites. Larger cells show more frequent

*Figure 4 continued on next page*

*Figure 4 continued*

multipolar outcomes (13.3%, 21.4%, 46.2%). Error bars indicate SEM from three experiments. (F) Diploid and haploid *cdc12-6 rsr1Δ* cells with normal (DLY15376, DLY9453) or double (DLY23308, DLY23302) the gene dosage of *CDC42, BEM1* and *CDC24* show that cell size is unaffected by polarity gene dosage. (G) The number of initial polarity sites does not vary systematically as a function of polarity gene dosage (strains as in F). (H) The percentage of two-budded cells within the subpopulation that established two initial polarity sites (strains as in F). Similar sized cells expressing more polarity proteins display more frequent multipolar outcomes (13.3% vs 29.4% in cycle two diploids, although not significant, Fisher's exact test p=0.25. 21.4% vs 68.4% in cycle three haploids, Fisher's exact test p=0.0094). Error bars indicate SEM from three experiments.

The online version of this article includes the following figure supplement(s) for figure 4:

**Figure supplement 1.** Polarity protein abundance as a function of strain and cell size.

**Figure supplement 2.** Effect of increased polarity gene dosage on the frequency of multi-polar outcomes.

The indirect substrate model is clarifying in its simplicity: lacking negative feedback, it demonstrates that the key to equalization is the presence of an indirect pathway to convert activator into substrate. As the minimalistic model already had a *direct* pathway to convert activator into substrate, this implies that it is important to introduce a delay in this conversion. Without this delay, the substrate generated from a peak is immediately available to be recruited back to the same peak. With a delay, the indirect substrate can migrate away from the center of the peak where it was generated before being converted to substrate, which can then feed a different peak. If this reasoning is correct, then the mobility of the indirect substrate should be critical for the model to yield equalization. To assess this, we examined how changing the diffusion constant of the indirect substrate would impact system behavior of a starting two-peak condition as the total amount of protein in the system was increased (*Figure 5H,I*). This analysis confirmed that the mobility of the indirect substrate must surpass a threshold in order to yield equalization, and showed that equalization became the favored outcome as that mobility increased (*Figure 5H*). Thus, equalization occurs when local production of a mobile indirect substrate by the larger peak drives a flux of substrate toward the smaller peak.

In addition, the analysis confirmed that the difference between the basal substrate level associated with each peak precisely predicted the outcome of the simulations (*Figure 5I*). When basal substrate was higher at the smaller peak, there was a flux of substrate toward the larger peak and the system displayed competition. When basal substrate was higher at the larger peak, there was a flux of substrate toward the smaller peak and the system displayed equalization. When basal substrate was very similar at the two peaks, the system displayed coexistence.

Our conclusions from analysis of the simple indirect substrate model can explain the outcomes from all of the models discussed above and in previous studies (*Figure 5—figure supplement 1*). All MCAS models can yield competition, but only models with indirect pathways to convert activator to substrate can yield equalization, and this occurs regardless of whether the models have negative feedback.

## Are multi-polar outcomes a result of equalization or coexistence?

Returning to the switch from unipolar to multipolar outcomes that we observed when yeast cells grew larger, we now have two potential explanations for this phenomenon: a slowing of competition to yield coexistence, or a switch to equalization. As the original impetus for the analysis of equalization stemmed from models that contained a phosphorylated and inhibited GEF, we wondered whether GEF phosphorylation might contribute to the outcome. We constructed a mutant strain in which the wild-type Cdc24 (GEF) was replaced with a non-phosphorylatable version, Cdc24$^{38A}$. This did not diminish multipolar outcomes: in fact, this strain made two-budded cells at a higher frequency than size-matched Cdc24-wild-type controls (of the second-cycle cells that established two initial sites, 65% of the Cdc24$^{38A}$ cells were two-budded compared to 12% of the Cdc24 cells).

Although phosphorylated Cdc24 is not required for multipolar outcomes, it remained possible that some other species (e.g. a mobile GAP or another species that acts as an indirect substrate) might be causing equalization. Equalization and competition are distinct behaviors in which initial polarity sites evolve in opposite directions. In contrast, coexistence is an intermediate behavior that could reflect either very slow competition or very slow equalization (*Figure 5H,I*). During equalization, two initially unequal peaks evolve toward two equal peaks. During coexistence, unequal peaks remain unequal. We asked which of these scenarios best accounts for the behavior of cytokinesis-defective yeast cells.

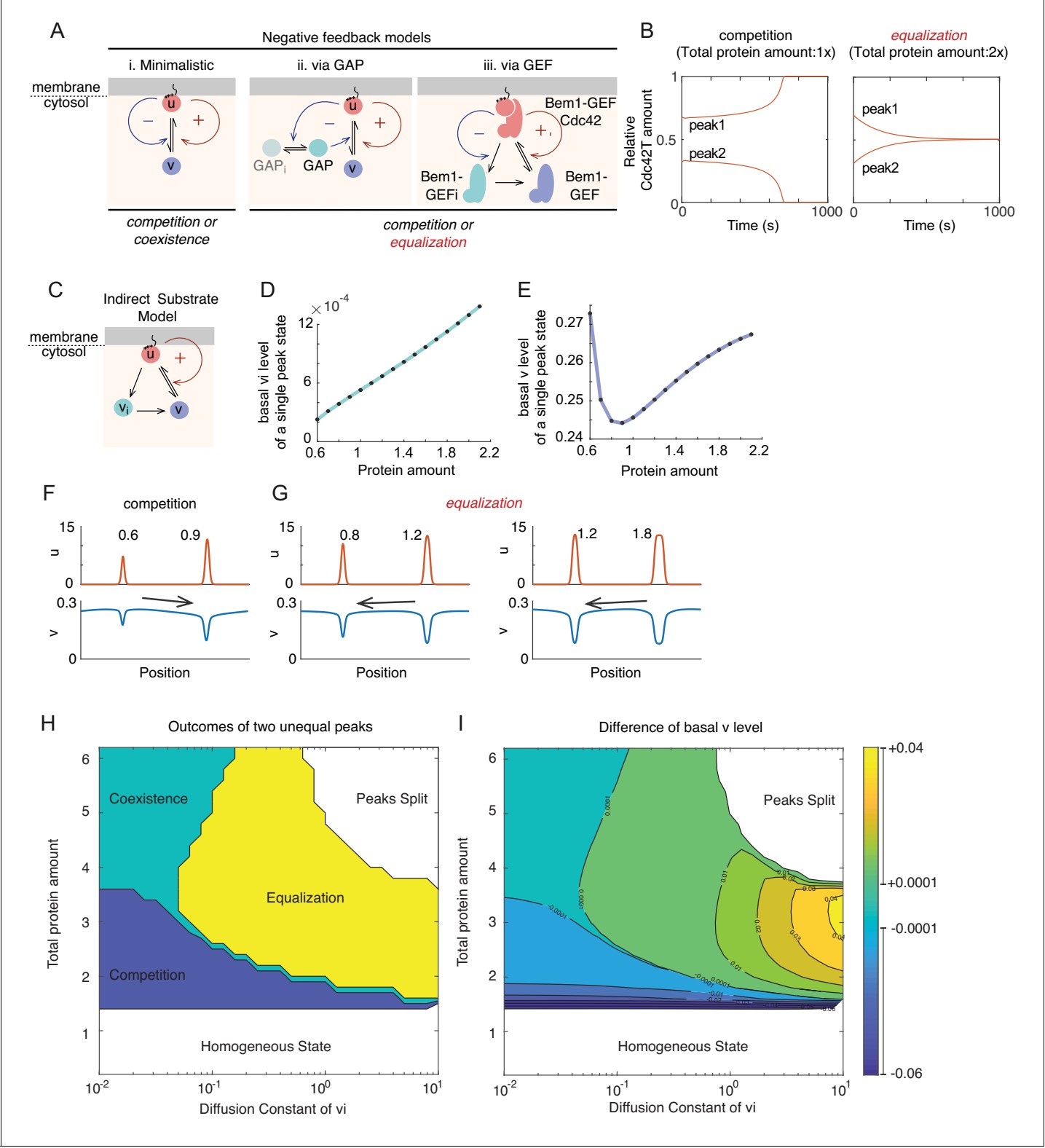

**Figure 5.** Basis for equalization in more complex models. (**A**) Schematic of models incorporating negative feedback. (**i**) In the minimalistic model, negative feedback adds a term in which *u* promotes conversion of *u* to *v* in a non-linear manner. (**ii**) This model incorporates conversion of an inactive GAP (GAPi) to an active GAP in a manner stimulated by *u*. The GAP promotes conversion of *u* to *v*, providing negative feedback. (**iii**) Simplified scheme of a mechanistic model where positive feedback occurs via recruitment of a substrate (Bem1-GEF) to become an activator (Bem1-GEF-Cdc42). Negative feedback occurs because the activator promotes inhibitory phosphorylation of the Bem1-GEF, generating inactive Bem1-GEFi. For details of the full

*Figure 5 continued on next page*

*Figure 5 continued*

model see *Figure 5—figure supplement 1F* and Materials and methods. (B) With the GEF negative feedback model, two peaks can compete (protein amount 0.5x, 1x) or equalize (protein amount 1x, 2x). (C) Schematic of the indirect substrate model. In addition to the reactions from minimalistic MCAS model in *Figure 1C,u* (red) can be converted into the indirect cytoplasmic substrate $v_i$ (teal), which itself can be converted to the substrate $v$ (blue). (D) The basal level of indirect substrate ($v_i$) increases as the amount of total protein in the system goes up. (E) The basal substrate level ($v$) first decreases but then increases as the amount of total protein in the system goes up. (F) Competition occurs when the larger peak depletes cytoplasmic substrates more than the smaller. Protein amount 0.6x:0.9x. (G) Equalization occurs when the smaller peak depletes cytoplasmic substrates more than the larger. Protein amount 0.8x:1.2x and 1.2x:1.8x. (H) Outcomes vary depending on the diffusion constant of $v_i$ and total protein amount. Simulations were initiated with two single peak steady states containing unequal total protein amounts 0.4:0.6. Outcomes were classified as follows: Competition, simulations evolve to a single peak steady state within t < 2000; Coexistence, unequal peaks remain unequal (±1%) within t < 2000; Equalization, simulations evolve to a two-equal-peak steady state within t < 2000; Peaks split, simulations evolve to split a starting peak into two smaller peaks; Homogeneous, simulations evolve to a homogeneous steady state. (I) Contour plot of the difference in basal $v$ concentration between the two initial peaks ($v_{P0.6}$ - $v_{P0.4}$). Contour lines are at 0.01 intervals except when basal levels are very close, when we indicate ±0.0001 contours. The starting difference in basal substrate between the peaks predicts the outcome.

The online version of this article includes the following figure supplement(s) for figure 5:

**Figure supplement 1.** Summary of model behaviors.

A difficulty in distinguishing equalization and competition behaviors in cells with wild-type Cdc24 is that negative feedback through Cdc24 phosphorylation can cause oscillation in the protein amount at each polarity peak, obscuring the underlying processes (*Howell et al., 2012*; *Kuo et al., 2014*). The $CDC24^{38A}$ strain short-circuits this negative feedback and does not show oscillations, allowing us to circumvent this complexity. Focusing on $CDC24^{38A}$ cells that had two starting polarity sites in the second cycle at 37°C, we tracked the total Bem1 fluorescence at each polarity site over time until the time of bud emergence. We quantified the fluorescence in each site as a fraction of the sum total in both sites, for each timepoint. Time-courses for 32 cells are shown in *Figure 6* and *Figure 6—figure supplement 1*. These cells exhibited a continuum of behaviors that could almost all be classified as competition (15 cells: the larger peak grew and the smaller one shrank, *Figure 6A*) or coexistence (14 cells: the relative Bem1 amount in each peak stayed approximately constant, *Figure 6C*). There were three cells in which quantification indicated that the larger peak shrank and the smaller one grew, consistent with equalization (*Figure 6D*). However, these instances of possible equalization were rare, and their interpretation is dependent on quantification of a single timepoint (the last timepoint before budding), which leaves us uncertain as to whether they represent equalization or coexistence. Interestingly, five of the cells with the slowest competition did not complete competition by the time of bud emergence, and went on to grow two buds (*Figure 6B*). This phenotype, which we call 'aborted competition', appears to reflect slow competition curtailed by budding. In aggregate, these data suggest that coexistence is the dominant reason for multipolar outcomes in yeast.

## Discussion

Previous studies on MCAS models applicable to cell polarity indicated that competition between polarity sites would yield unipolar final states, but that the timescale of competition would slow as the amount of polarity proteins in the system increased, potentially yielding coexistence of polarity sites and hence multipolar outcomes (*Brauns et al., 2020a*; *Chiou et al., 2018*; *Goryachev and Leda, 2020*). Our findings expand on this work in three ways. First, we show experimentally that yeast cells behave as predicted by the models, producing multipolar outcomes as the amount of polarity proteins increases. Second, we provide an explanation for a phenomenon called equalization, seen in more complex MCAS models, which can also produce multipolar outcomes. And third, we documented the dynamics of polarity sites in yeast cells, supporting coexistence as the dominant mechanism for multi-polar outcomes.

### Yeast cells switch from unipolar to multipolar outcomes as polarity protein abundance is increased

Using conditional cytokinesis-defective yeast mutants, we found that larger cells produced progressively higher numbers of buds. Increasing numbers of polarity sites with cell size is predicted by

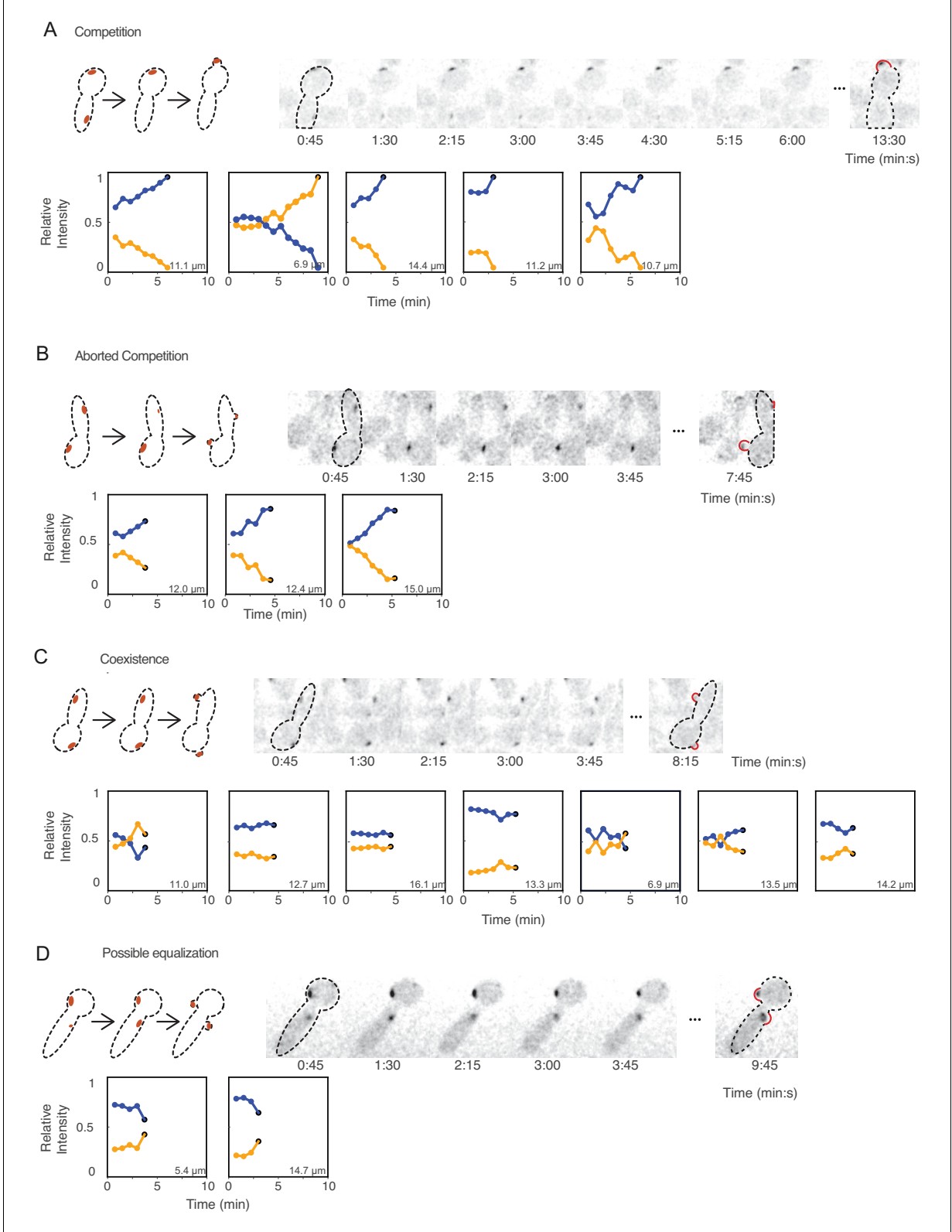

**Figure 6.** Competition, coexistence, and equalization in yeast. Cells that exhibit (**A**) competition, (**B**) aborted competition, (**C**) coexistence, or (**D**) possible equalization. Cartoons depict the dynamics of polarity sites (red) in the accompanying montages. Red bulges (right) indicate buds, which sometimes emerge away from the focal plane (note that polarity sites in small buds were always at the bud tip, but the appearance in 2D maximum projections may not convey that when buds grow away from the focal plane). The amounts of Bem1 in each site were quantified from sum intensity

*Figure 6 continued on next page*

*Figure 6 continued*

traces from z-stacks of *cdc12-6 rsr1Δ CDC24$^{38A}$* cells (DLY21100) that had two initial polarity sites. We plotted the relative amounts of Bem1 at the two polarity sites from when both sites had grown until the time of budding, with blue and orange dots representing the initially more (blue) or less (orange) intense polarity sites. Polarity sites that eventually led to bud-emergence are indicated by black dots at the last time point. Inset numbers denote the distance between the two initial polarity sites. Timelapse montages show the first cell in each category. Additional traces are shown in *Figure 6—figure supplement 1*.

The online version of this article includes the following figure supplement(s) for figure 6:

**Figure supplement 1.** Competition, coexistence, and equalization outcomes are not correlated with the distance between initial polarity sites.

different classes of simple mathematical pattern-formation models relevant to polarity establishment. The models provide different explanations (intrinsic length scale, coexistence, and equalization) for the switch from unipolar to multipolar outcomes.

Classical (not mass-conserved) turing models exhibit a characteristic length scale, such that local peaks of an activator form spontaneously at spatial intervals corresponding to this length scale. Cells develop one or more peaks depending on how large the cell is compared to the characteristic length scale (*Cornwall Scoones et al., 2020*; *Meinhardt, 2008*).

Mass-conserved activator-substrate (MCAS) models can develop variable numbers of initial activator peaks, but as substrate is depleted from the cytoplasm the peaks compete with each other. Eventually, competition leads to a single-peak steady state, but the timescale of competition slows dramatically as the total abundance of activator/substrate in the system is increased (*Brauns et al., 2020a*; *Chiou et al., 2018*; *Ishihara et al., 2007*; *Otsuji et al., 2010*), yielding coexistence on biologically relevant timescales.

More complex MCAS models can switch from a regime exhibiting competition (where a larger peak grows faster than a smaller peak) to a regime exhibiting equalization (where a smaller peak grows faster than a larger peak) as parameters change (*Howell et al., 2012*). The basis for this behavior is discussed in more detail below. For now, we note that a switch from competition to equalization as cells become larger could also explain the observed switch from unipolar to multipolar outcomes.

The number of initial polarity sites in our cells increased with increasing cell length, reminiscent of classical Turing models. However, there was no obvious preferred length scale for the distance between sites, as would be expected from classical Turing systems. A potential explanation for the increase in initial sites stems from the observation that polarity locations were non-random, with a preference for cell tips. For the geometry of our cytokinesis-defective cells, the number of tips correlates with cell length, because longer cells arise from the formation of additional buds. Thus, the number of initial polarity sites may reflect the specifics of our experimental system rather than a feature of classical Turing systems.

Those cells that did form more than one initial polarity peak often exhibited competition between peaks. Competition emerges as a consequence of mass conservation, and suggests that MCAS models provide the explanation that best fits the behavior of the yeast system. Additional copies of genes encoding polarity proteins led to a marked increase in the frequency of multi-polar outcomes with no change in cell size. Tracking a polarity marker in time-lapse imaging, we found two predominant behaviors among large yeast cells that generated two initial polarity sites: one subset exhibited competition between sites to yield a single bud, while the other exhibited apparent coexistence between sites yielding two buds. These findings strongly support the idea that multipolar outcomes in this system arise due to a slowing in the timescale of competition.

## Other features of competition in cytokinesis-defective cells

A subset of our cells exhibit aborted competition, which failed go to completion before budding and gave rise to two buds with stable polarity sites. This behavior is at odds with the predictions of MCAS models, in which competition accelerates as the amount of activator in the peaks becomes more uneven and always goes to completion. This discrepancy may indicate that some aspect of the polarity circuit changes at around the time of bud emergence, reducing the efficacy of competition. Cell cycle control by the cyclin/CDK system provides one plausible candidate regulator that could

prompt such a change in polarity circuit behavior (*Knaus et al., 2007*; *Moran et al., 2019*; *Sopko et al., 2007*; *Witte et al., 2017*).

While competition could occur between peaks in the same or different lobes of cytokinesis-defective cells, all of the multibudded cells we observed had buds emanating from different cell lobes. This observation suggests that competition is more effective within a lobe than between lobes. A possible basis for this effect is that the geometrical narrowing at the neck of multi-lobed cells slightly retards diffusional communication between lobes.

## Saturation in multi-component MCAS models

Analyses of minimalistic two-component MCAS models demonstrated that competition between activator peaks was inevitable, and that the timescale of competition was determined by a single dominant factor, which we refer to as saturation (*Brauns et al., 2020a*; *Chiou et al., 2018*; *Jacobs et al., 2019*; *Otsuji et al., 2010*). When activator concentration in two or more peaks approaches a saturation point set by system parameters, competition slows dramatically, allowing coexistence of the peaks on biologically relevant timescales. We show that these findings from minimalistic one-activator, one-substrate models hold in a more complex and realistic model of the yeast polarity circuit, with two activators, two substrates, and two intermediate species, with some additional complexity discussed below.

In minimalistic MCAS models, positive feedback ensures that addition of more substrate/activator to the system results in conversion of more substrate to activator. Due to positive feedback, the concentration of substrate is depleted below the level obtained with less substrate/activator in the system at steady state. In addition, the concentration profile of activator in the peak changes in a characteristic way as the local activator concentration approaches the saturation point, flattening from a sharp peak to a mesa. In the multi-component model, substrate depletion and the activator concentration profile can vary for different protein species. Depending on the relative amounts of the different species, different substrates may become limiting. For the limiting species, addition of more protein will generally lead to depletion of the cytoplasmic substrate, similar to saturation in the minimalistic models. However, other (non-limiting) species can display increasing cytoplasmic substrate concentration as more protein is added. Moreover, addition of one species can lead to a switch in the identity of the limiting species. As with minimalistic MCAS models, the timescale of competition in the mechanistic multi-component model slows as more protein is added to the system, in a manner consistent with saturation of the limiting species.

## Equalization in multi-component MCAS models

Unlike minimalistic MCAS models, previous research has shown that more complex models incorporating negative feedback can yield equalization of polarity peaks in some parameter regimes (*Howell et al., 2012*; *Jacobs et al., 2019*). Local negative feedback provides an intuitive rationale for equalization: by penalizing a larger peak more than a smaller peak, a localized negative feedback loop could switch the competitive advantage toward the smaller peak. However, we found that negative feedback was neither necessary nor sufficient to produce equalization. Instead, the key to equalization is the existence of additional species that provide indirect pathways to convert an activator into a substrate.

The key features of an additional species that enable equalization are as follows. First, the new species is produced from (or by) an activator. Second, the new species produces substrate. Third, the new species must be more mobile than the activator. In combination, these features create a pathway whereby activator from a peak is converted to a new species that diffuses away from the peak before it generates substrate, which can then be used to 'feed' a different peak. A larger activator peak produces more of the new species than a smaller peak, yielding a flux of mobile species from the larger peak to the smaller peak that can produce equalization.

The simplest model containing an indirect pathway from activator to substrate is a three-species model with an activator, a substrate, and an indirect substrate. Using this model, we characterized the full spectrum of behaviors as a function of the mobility of the indirect substrate (*Figure 5I*). When the indirect substrate diffused slowly (comparable to the activator), the system exclusively displayed competition or coexistence, indicating that rapid mobility of the indirect substrate is essential for equalization. At the other extreme, when the indirect substrate diffused much faster than the

substrate, the system exhibited classical (not mass conserved) Turing behavior, generating peaks separated by a characteristic wavelength. We note that in the limit of infinite diffusion of the indirect substrate, this mass-conserved system would resemble some models that do not assume mass conservation (*Brauns et al., 2020a*; *Jacobs et al., 2019*). In that limit, the indirect substrate concentration is uniform, so that its conversion into substrate becomes equivalent to a simple 'substrate synthesis' term. As indirect substrate is produced from activator, that process becomes equivalent to a simple 'activator degradation' term. Thus, in that limit the system behaves as if synthesis and degradation were allowed, so each peak's activator concentration profile reflects a local steady state between synthesis and degradation. The peaks are equal because they share the same synthesis and degradation parameters. Finally, when the mobility of the indirect substrate was comparable to that of the substrate (i.e. in between the extremes considered above), the system exhibited competition or equalization depending on the total abundance of polarity proteins.

Do yeast cells contain indirect pathways from activator to substrate that could enable equalization? Past work suggests several candidates for such pathways. For example, GTP-Cdc42 promotes inhibitory phosphorylation of its GEF (*Kuo et al., 2014*), and a model in which GTP-Cdc42 promotes GEF phosphorylation to make GEFi (new species) that can then be dephosphorylated to make active GEF (substrate) can yield equalization (*Figure 5B*). Similarly, a Cdc42-directed GAP is locally concentrated/activated by proteins downstream of Cdc42 (*Okada et al., 2013*), and a model in which GTP-Cdc42 activates a GAP (new species) that can then convert GTP-Cdc42 to GDP-Cdc42 (substrate) can also yield equalization (*Jacobs et al., 2019*). Moreover, the yeast polarity circuit has several other species that could similarly provide indirect pathways from activator to substrate. For example, a multi-component complex containing GTP-Cdc42 and bound effector, scaffold, and GEF proteins (activator) can dissociate to yield individual components (GEFs, scaffolds, effectors) that are absent in minimalistic models. Only when these proteins re-associate to form cytoplasmic GEF-scaffold-effector complexes (substrate) can they be recruited back to bind GTP-Cdc42 at the polarity site. Thus, there are multiple plausible candidates that could enable equalization in the yeast polarity circuit. However, we detect very few cases of possible equalization in the cells we analyzed, suggesting that the physiological parameter space in which this circuit operates is not prone to equalization.

## Implications for other systems

Turing-type systems have enormous and well-appreciated potential to generate biologically useful patterns (*Meinhardt, 2008*; *Meinhardt and Gierer, 2000*). However, they can also display chaotic outcomes that change dramatically with small differences in starting conditions or parameter values (*Pearson, 1993*). Thus, biological circuits that exploit such reaction-diffusion systems are likely to employ only a small subset of such circuits that robustly produce reliable outcomes. MCAS circuits are now recognized to produce desirable outcomes for morphogenetic biological systems, including polarization, in a robust manner (*Goryachev and Leda, 2017*). Moreover, recent advances have provided insight into how such systems can be tuned to produce unipolar or multipolar outcomes (*Chiou et al., 2018*; *Goryachev and Leda, 2020*; *Halatek et al., 2018*). One property that can effectively control the number of polarity sites that form is cell size (*Cornwall Scoones et al., 2020*). For example, there is a minimum size below which MCAS circuits are unable to polarize at all, and recent work indicates that the decrease in cell size that occurs during worm embryogenesis causes a switch from asymmetric cell division of polarized large cells to symmetric division of unpolarized small cells (*Hubatsch et al., 2019*).

Our findings show that yeast cells can switch from having a single polarity site to two or more polarity sites as cell size increases. The *Saccharomyces* polarity circuit has presumably been evolutionarily selected to produce uni-polar outcomes, which are beneficial during budding and mating in this genus. However, this polarity circuit is highly conserved among ascomycetes that display other growth modes (*Bendezú et al., 2015*; *Lamas et al., 2020*). *Schizosaccharomyces pombe* naturally switch from uni-polar to bi-polar growth during each cell cycle (*Grallert et al., 2013*; *Martin and Chang, 2005*). *Ashbya gossypii*, which evolved relatively recently from a common ancestor with *S. cerevisiae*, form branching hyphae that exhibit increasing number of polarity sites as each cell grows (*Knechtle et al., 2003*; *Schmitz and Philippsen, 2011*). The correlation between cell size and number of polarity sites in these systems suggests that, as proposed for MCAS models, cell size and the associated higher abundance of polarity factors may trigger the increase in number of polarity sites. Most remarkably, yeast cells of *Aureobasidium sp.* generate variable numbers of buds

simultaneously (*Mitchison-Field et al., 2019*), a capacity shared with the much more distantly related zygomycete *Mucor circinelloides* (*Lee et al., 2013*). This phenotypic diversity may be enabled by a polarity circuit that allows a switch between competition, coexistence, and equalization behaviors in response to appropriate tuning of parameter values. Similar principles may apply to other systems where activator species produce robustly tunable numbers of polarity sites.

# Materials and methods

## Yeast strains

All yeast strains (*Table 1*) are in the YEF473 background (*his3-Δ200; leu2-Δ1; lys2-801amber; trp1-Δ63; ura3-52*)(*Bi and Pringle, 1996*). The *cdc12-6* mutation in the YEF473 background was a gift from John Pringle (Stanford University). The *rsr1* deletion (*Schenkman et al., 2002*), *GAL4BD-hER-VP16* construct (*Takahashi and Pryciak, 2008*), and *CDC24[38A]* mutation (*Kuo et al., 2014*; *Wai et al., 2009*) were described previously, as was tagging at the endogenous loci for the fluorescent probes *BEM1-GFP* (*Kozubowski et al., 2008*), *BEM1-tdTomato* (*Howell et al., 2012*), *CDC3-mCherry* (*Howell et al., 2009*), *CLA4-GFP* (*Wild et al., 2004*), *HTB2-mCherry*, and *WHI5-GFP* (*Doncic et al., 2011*). Standard yeast genetic procedures were used to generate all of the strains.

**Table 1.** Strains.

| Strain | Relevant genotype | Source |
|---|---|---|
| DLY9453 | *a; cdc12-6; rsr1::HIS3; BEM1-GFP:LEU2* | This study |
| DLY9455 | *a; cdc12-6; BEM1-GFP:LEU2* | This study |
| DLY13030 | *a; cdc42::TRP1; rsr1::TRP1; URA3:GFP-CDC42* | This study |
| DLY15376 | *a/α; cdc12-6/cdc12-6; rsr1::HIS3/rsr1::HIS3; BEM1-GFP:LEU2/BEM1-GFP:LEU2* | This study |
| DLY16767 | *a; cdc12-6; rsr1::TRP1; BEM1-GFP:LEU2; CDC24[38A]* | This study |
| DLY20240 | *a; cdc12-6; BEM1-GFP:LEU2; HTB2-mCherry:nat[R]* | This study |
| DLY20569 | *a/α; cdc12-6/cdc12-6; BEM1-GFP:LEU2/BEM1-GFP:LEU2* | This study |
| DLY21100 | *a/α; cdc12-6/cdc12-6; BEM1-GFP:LEU2/BEM1-GFP:LEU2; CDC24[38A]/CDC2438[38A]* | This study |
| DLY22875 | *a; cdc12-6; WHI5-GFP:SpHIS5; BEM1-GFP:LEU2; pTEF1-PSR1-mCherry-tADH1:LEU2; rsr1::TRP1* | This study |
| DLY22887 | *a/matαΔ::nat[R]; cdc12-6/cdc12-6; BEM1-GFP:LEU2/BEM1-GFP:LEU2* | This study |
| DLY22915 | *a/α; WHI5-GFP:SpHIS5/WHI5; rsr1::TRP1/RSR1; BEM1-tdTomato:HIS3/BEM1-tdTomato:HIS3; pGAL1-IQG1:LEU2/pGAL1-IQG1:LEU2; pTEF1-PSR1-GFP-tADH1:LEU2/leu2; Gal4BD-hER-VP16:URA3/Gal4BD-hER-VP16:URA3* | This study |
| DLY22920 | *a/α; cdc12-6/cdc12-6; WHI5-GFP:SpHIS5/WHI5-GFP:SpHIS5; BEM1-tdTomato:HIS3/BEM1-tdTomato:HIS3* | This study |
| DLY22957 | *a/α; cdc12-6/cdc12-6; BEM1-tdTomato::HIS3/BEM1; pTEF1-GFP:LEU2/pTEF1-GFP:LEU2* | This study |
| DLY22980 | *α; rsr1::TRP1; BEM1-GFP:LEU2; CDC24-HA:kan[R]* | This study |
| DLY22993 | *α; cdc12-6; BEM1-GFP:LEU2; CDC24-HA:kan[R]* | This study |
| DLY23302 | *a; cdc12-6; CDC24-HA:kan[R]; BEM1-GFP:TRP1:BEM1-GFP:LEU2; ura3::CDC24-HA:URA3; CDC42:TRP1:CDC42; rsr1::TRP1* | This study |
| DLY23308 | *a/α; cdc12-6/cdc12-6; CDC24-HA:kan[R]/CDC24-HA:kan[R]; BEM1-GFP:TRP1:BEM1-GFP:LEU2/BEM1-GFP:TRP1:BEM1-GFP:LEU2; ura3::CDC24-HA:URA3/ura3::CDC24-HA:URA3; CDC42:TRP1:CDC42/CDC42:TRP1:CDC42; rsr1::TRP1/rsr1::TRP1* | This study |
| DLY23359 | *a/α; cdc12-6/cdc12-6; CLA4-GFP::HIS3/CLA4; BEM1-tdTomato::HIS3/BEM1* | This study |
| DLY23269 | *α; cdc12-6; rsr1::TRP1; BEM1-GFP:TRP1:BEM1-GFP:LEU2; CDC24-HA:kan[R]; ura3::CDC24-HA:URA3; CDC42:TRP1:CDC42* | This study |
| DLY23829 | *a; cdc12-6; rsr1::HIS3; BEM1-GFP:TRP1:BEM1-GFP:LEU2* | This study |
| DLY23830 | *a; cdc12-6; rsr1::TRP1; BEM1-GFP:LEU2; CDC24-HA:kan[R]; ura3::CDC24-HA:URA3* | This study |
| DLY23831 | *a; cdc12-6; rsr1::HIS3; BEM1-GFP:LEU2; CDC42:TRP1:CDC42* | This study |
| DLY23832 | *a; cdc12-6; rsr1::TRP1; BEM1-GFP:LEU2; ura3::CDC24-HA:URA3; CDC42:TRP1:CDC42* | This study |
| DLY23833 | *α; cdc12-6; rsr1::HIS3; BEM1-GFP:TRP1:BEM1-GFP:LEU2; CDC24-HA:KAN; CDC42:TRP1:CDC42* | This study |
| DLY23843 | *α; cdc12-6; rsr1::HIS3; BEM1-GFP:TRP1:BEM1-GFP:LEU2; CDC24-HA:KAN; ura3::CDC24-HA:URA3* | This study |

To generate a strain with regulatable expression of *IQG1*, the first 500 bp of the *IQG1* open reading frame were amplified by PCR and cloned downstream of the *GAL1* promoter in YIpG2 (*Richardson et al., 1989*) to generate DLB2126. Digestion at the unique *Nhe*I site targets integration of this construct at *IQG1*, making Iqg1 expression galactose-dependent and shut off on glucose media. To introduce a 3xHA epitope tag at the C-terminus of *CDC24*, we used a pFA6-series plasmid template and the PCR-based one-step replacement method (*Longtine et al., 1998*). To delete the MATα locus, we used a pFA6-series plasmid template and the PCR-based one-step replacement method (*Longtine et al., 1998*) to replace a part of the locus inactivating the divergent α1 and α2 genes while leaving the surrounding genes intact. In a haploid, this deletion converts an α mating type to an **a** mating type. In a diploid, this deletion converts the strain to **a** mating type.

To label the plasma membrane, we expressed a fusion between the N-terminal 28 residues of Psr1 and GFP. The Psr1 N-terminal fragment is myristoylated and doubly palmitoylated, targeting GFP to the plasma membrane (*Siniossoglou et al., 2000*). The construct was cloned between the *TEF1* promoter and *ADH1* terminator sequences in a pRS305 (*Sikorski and Hieter, 1989*) backbone, generating plasmid DLB4206. Digestion at the unique *Ppu*MI targets integration at the *LEU2* locus.

To express an extra copy of *CDC42*, the *CDC42* gene (open-reading frame plus 500 bp upstream and 250 bp downstream) was cloned into the integrating plasmids pRS304 and pRS306 (*Sikorski and Hieter, 1989*), generating plasmids DLB3904 and DLB4115 respectively. Digestion of DLB3904 at the unique *Sty*I site was used to target integration of the *TRP1*-marked plasmid at *CDC42*. To express an extra copy of *CDC24*, the *CDC24-3HA* gene (open reading frame plus upstream and downstream sequence) was cloned into the integrating plasmid pRS306 (*Sikorski and Hieter, 1989*), generating plasmid DLB4134. Digestion at the unique *Pst*I site was used to target integration of the plasmid at *URA3*. To express an extra copy of *BEM1-GFP*, the *BEM1-GFP* gene (open reading frame plus upstream and downstream sequence) was cloned into the integrating plasmid pRS304 (*Sikorski and Hieter, 1989*), generating plasmid DLB2997. Digestion at the unique *Bam*HI site was used to target integration of the plasmid at *BEM1*.

## Cell growth, hydroxyurea treatment, and timelapse imaging conditions

Cells were grown in liquid complete synthetic media (CSM, MP Biomedicals) with 2% dextrose at 24° C overnight until they reached log phase ($5 \times 10^6$ cells/mL). *cdc12-6* cultures were shifted to 37°C and treated with 200 mM hydroxyurea (Sigma) for 1 hr to protect cells from subsequent phototoxicity during imaging (*Howell et al., 2012*). Cells were pelleted, washed with and released into fresh media at 37°C for an additional 1 hr (for imaging of the second cell cycle) or 3 hr (for imaging of the third cell cycle). Cells were then harvested by centrifugation and mounted on a 37°C slab composed of CSM solidified with 2% agarose (Denville Scientific, Inc) prior to imaging.

For *Figure 2A*, the cells were imaged at 37°C on an Axio Observer.Z1 (Zeiss) with Pecon XL S1 incubator and control modules, a X-CITE 120XL metal halide fluorescence light source, and a 100x/1.46 (Oil) Plan Apochromat objective controlled by MetaMorph 7.8 (Universal Imaging). Images were captured with a Photometrics Evolve back-thinned EM-CCD camera. The fluorescence light source was set to 50% of the maximal output with a 2% ND filter. An EM-Gain of 750 and 200 ms exposure was set for the red channel (HTB2-mCherry, not shown), and an EM-Gain of 100 and 20 ms exposure was set for the Differential interference contrast (DIC) channel.

Other videos and images were acquired with an Andor XD revolution spinning disk confocal microscope (Olympus) with a Yokogawa CsuX-1 5000 rpm disk unit and a 100x/1.4 U PlanSApo oil-immersion objective controlled by MetaMorph 7.8. 20 Z-stacks of 0.5 μm z-step were captured at 45 s intervals with Andor Ixon3 897 512 EMCCD camera (Andor Technology).

The maximal power for the 488 nm laser varied between 1.60 mW and 3.26 mW, and the maximal power for the 561 nm laser varied between 1.19 mW and 1.68 mW. We adjusted the illumination to 6–8% for the 488 nm channel and 8–10% for the 561 nm channel to provide a more consistent sample illumination. An EM-Gain of 200 and exposures of 250 ms were used.

Fluorescent images were deconvolved with SVI Huygens Deconvolution (Scientific Volume Imaging) and analyzed using Fiji (*Schindelin et al., 2012*). For deconvolution, a signal to noise ratio of 3 was used for Confocal images. Only cells that were not connected to neighboring cells were used for quantification to avoid cell pairs that might be connected from the previous cell cycle.

## Cell fixation and membrane staining

To score the number of buds at the second cell cycle in *Figure 2D*, cells were grown overnight at 24°C and log phase cultures ($10^7$ cells/mL) were shifted to 37°C for 4 hr. One mL cell culture was then harvested, spun down, and resuspended in 100 µL ice cold 10 µM FM4-64fx in water (Thermofisher Scientific) on ice. After 1 min staining, 1 mL ice cold 4% paraformaldehyde was added and the mixture was incubated on ice for 10 min. The cells were then washed twice with phosphate-buffered saline (PBS) and stored at 4°C. Images were then taken, and only cells that were budding from two connected compartments were counted.

## Photo-bleaching

Photo-bleaching experiments were conducted on a DeltaVision Elite Deconvolution Microscope (Applied Precision) with a 100x/1.40 oil UPLSAPO100 × 0 1-U2B836 WD objective controlled by SoftWoRx 6.1 (Softworx Inc). Images were captured with a Coolsnap HQ2 high resolution CCD camera. Photobleaching experiments were conducted on budded cycle two *cdc12-6* cells expressing cytoplasmic GFP. The bleaching 488 nm laser was used for 5 ms at 20% of maximal intensity. Cells were imaged with 50 ms exposure time, and 2 × 2 binning for three images before bleach and 15 images after bleach. Imaging interval was set automatically by the software assuming 1 s half-time.

Images were analyzed using Fiji and MATLAB (Mathworks). Fluorescence signal was averaged within a 3 µm diameter circular area at the bleach site and at sites in the mother and daughter compartments equidistant to the bleach site, and normalized with an unbleached cell and the background fluorescence nearby using the formula:

$$I_{normalized} = (I_{raw} - I_{background})/(I_{unbleached} - I_{background})$$

Normalized intensity at the mother site was fit to an exponential decay $ae^{-kt} + c$, normalized intensity at the bleach site was fit to an exponential recovery $-ae^{-kt} + c$, and normalized intensity at the daughter site was fit to a linear combination of the two $ae^{-kt} + be^{ct} + d$. The recovery half-time can then be calculated by $T_{1/2} = \ln(2)/k$.

## Simulated Photo-bleaching in 3D cells

The three-dimensional geometry of a typical second-cycle *cdc12-6* cell was modeled by the closest point method described in *Ramirez et al., 2015*. The cell shape was designated as the combination of a 6 µm diameter sphere and an ellipsoid with length 6 µm and width 2 µm, partially overlapped to create a neck of 2 µm diameter. The shape of the cell was modeled in Cartesian coordinates with the boundary of the cell interpolated with the closest grid points. The closest points were implemented with C++, and the main diffusion code was simulated by the implicit Euler method in MATLAB. The bleach was incorporated in the initial condition as a cylinder of 1 µm diameter and zero intensity. 'Fluorescence intensities' were the measured from the sum of z-stacks to mimic non-deconvolved microscopy images from the DeltaVision microscope.

## Immunoblotting

Cells were grown overnight in YEPD at 24°C to mid-log phase. Where indicated, cultures were shifted to 37°C for 4 hr prior to TCA precipitation. For all samples,~$10^7$ cells were collected via centrifugation. Pellets were resuspended in 225 µl of cold pronase buffer (25 mM Tris-HCl, pH 7.5, 1.4 M sorbitol, 20 mM $NaN_3$, 2 mM $MgCl_2$) and 48 µl of cold TCA (100% wt/vol; Sigma-Aldrich) before being frozen at −80°C. Samples were thawed on ice and cells were homogenized by vortexing with sterile acid-washed glass beads at 4°C for 10 min. Lysates were collected and the beads were washed with TCA (5% wt/vol) to collect remaining lysate. Precipitated proteins were pelleted by centrifugation at 4°C for 10 min. Pellets were resuspended in Thorner sample buffer (40 mM Tris-HCl, pH 6.8, 8 M urea, 5% SDS, 143 mM β-mercaptoethanol, 0.1 mM EDTA, 0.4 mg/ml bromophenol blue) and any remaining TCA was neutralized by adding 2 M Tris-HCl, pH 8.0.

Samples were heated at 95°C for 5 min prior to loading on 12.5% polyacrylamide gels. After electrophoresis, proteins were transferred to nitrocellulose membranes. Membranes were blocked with 3% nonfat dry milk in phosphate-buffered saline with 0.1% Tween-20 (PBST). Blots were incubated in blocking buffer with monoclonal mouse anti-GFP antibodies (Roche) at 1:1000 dilution, monoclonal mouse anti-HA antibodies (Roche) at 1:1000 dilution, or monoclonal mouse anti-Cdc42 antibodies

(*Wu and Brennwald, 2010*) at 1:500 dilution. After multiple washes with PBST, blots were incubated in blocking buffer with 0.01% SDS and fluorophore-conjugated secondary anti-mouse antibodies (IRDye 800CW goat anti-mouse IgG, LI-COR) at 1:10,000 dilution. Following multiple washes with PBST plus 0.01% SDS, blots were visualized and quantified using the ODYSSEY imaging system (LI-COR). Two percent Ponceau S solution was used to detect total protein in each blot and ImageJ was used to quantify total protein in each lane. For a given blot, the signal for each detected band was scaled according to the total protein measured in its corresponding lane. These scaled intensities were then normalized to the wild-type signal for that blot.

## Polarity models

We performed simulations with four polarity models in this study: a minimalistic mass-conserved activator-substrate (MCAS) model, two mechanistic models of the yeast polarity circuit with or without the negative feedback via GEF phosphorylation, and an extension of the minimalistic MCAS model that incorporates an indirect substrate.

The minimalistic MCAS model considers the concentrations of two interconvertible forms of a protein (activator and substrate: u, v) in one spatial dimension (*Figure 1C*; *Chiou et al., 2018*). The protein can diffuse and convert between the two forms but is not synthesized or degraded:

$$\frac{\partial u}{\partial t} = au^2v - bu + D_m\frac{\partial^2 u}{\partial x^2}$$

$$\frac{\partial u}{\partial t} = au^2v - bu + D_m\frac{\partial^2 u}{\partial x^2}$$

$$\frac{\partial v}{\partial t} = -au^2v + bu + D_v\frac{\partial^2 v}{\partial x^2}$$

u enhances the conversion of v into more u through an implicit positive feedback loop modeled by the quadratic term au$^2$v. u converts back to v in a first order process. u diffuses slowly relative to v. The parameters are:

| Description | Parameter | Value | Reference |
|---|---|---|---|
| u → v | a | 1 | *Chiou et al., 2018* |
| v → u | b | 1 | *Chiou et al., 2018* |
| Diffusion constant of u | D$_m$ | 0.01 | *Chiou et al., 2018* |
| Diffusion constant of v | D$_c$ | 1 | *Chiou et al., 2018* |

The indirect-substrate model is similar to the minimalistic model except for the inclusion of a new species, the indirect-substrate v$_i$ (*Figure 5C*). u converts to v$_i$ in a first-order process, and v$_i$ converts to v in a first-order process. The differences from the minimalistic MCAS model are highlighted in bold:

$$\frac{\partial u}{\partial t} = au^2v - bu - \mathbf{cu} + D_m\frac{\partial^2 u}{\partial x^2}$$

$$\frac{\partial v}{\partial t} = -au^2v + bu + \mathbf{dv_i} + D_v\frac{\partial^2 v}{\partial x^2}$$

$$\frac{\partial \mathbf{v_i}}{\partial \mathbf{t}} = \mathbf{cu} - \mathbf{dv_i} + \mathbf{D_{vi}}\frac{\partial^2 \mathbf{v_i}}{\partial \mathbf{x^2}}$$

| Description | Parameter | Value | Reference |
|---|---|---|---|

*Continued on next page*

*Continued*

| Description | Parameter | Value | Reference |
|---|---|---|---|
| u → v | a | 1 | *Chiou et al., 2018* |
| v → u | b | 1 | *Chiou et al., 2018* |
| u → $v_i$ | c | 0.01 | This study |
| $v_i$→ v | d | 1 | This study |
| Diffusion constant of u | $D_m$ | 0.01 | *Chiou et al., 2018* |
| Diffusion constant of v | $D_c$ | 1 | *Chiou et al., 2018* |
| Diffusion constant of v | $D_{vi}$ | 1 | This study |

The mechanistic positive feedback model (*Wu et al., 2015*) is based on reactions assumed to occur with first order kinetics that account for various interconversions of Cdc42 and PAK-Bem1-GEF complexes. Cdc42 can interconvert between active GTP-bound (Cdc42T) and inactive GDP-bound (Cdc42D) states. Activation is catalyzed by GEF at the membrane, while inactivation is catalyzed by an implicit GAP. GDP-Cdc42 can also exchange between membrane ($Cdc42D_m$) and cytoplasmic ($Cdc42D_c$) forms (in cells this is catalyzed by GDP-dissociation Inhibitor or GDI). The PAK-Bem1-GEF complex (here called BemGEF) is considered as a single species following the analysis of *Goryachev and Pokhilko, 2008*, who showed that separating the complex into distinct components did not affect the qualitative behavior of the system in the parameter ranges they considered (although we note that given the potential for separate PAK, Bem1, or GEF species to act as indirect substrates it is possible that considering the species separately would yield different outcomes in some parameter regimes). BemGEF can exchange between membrane ($BemGEF_m$) and cytoplasmic ($BemGEF_c$) forms, and in all cases membrane species diffuse much less than cytoplasmic species. Positive feedback occurs due to reversible binding of BemGEF to Cdc42T, generating the complex BemGEF42 at the membrane. This leads to accumulation of GEF at sites with elevated GTP-Cdc42, which promotes local activation of more Cdc42. These reactions are modeled as:

$$\frac{\partial}{\partial t}Cdc42T = (k_{2a}BemGEF_m + k_3BemGEF42) \cdot Cdc42D_m - (k_{2b} + k_{4a}BemGEF_m + k_7BemGEF_c) \cdot Cdc42T + k_{4b}BemGEF42 + D_m\Delta Cdc42T$$

$$\frac{\partial}{\partial t}Cdc42D_m = k_{2b}Cdc42T - (k_{2a}BemGEF_m + k_3BemGEF42) \cdot Cdc42D_m - k_{5b}Cdc42D_m + k_{5a}Cdc42D_c + D_m\Delta Cdc42D_m$$

$$\frac{\partial}{\partial t}BemGEF42 = (k_{4a}BemGEF_m + k_7BemGEF_c) \cdot Cdc42T - k_{4b}BemGEF42 + D_m\Delta BemGEF42$$

$$\frac{\partial}{\partial t}BemGEF_m = k_{1a}BemGEF_c - k_{1b}BemGEF_m + k_{4b}BemGEF42 - k_{4a}BemGEF_m \cdot Cdc42T + D_m\Delta BemGEF_m$$

$$\frac{\partial}{\partial t}Cdc42D_c = \eta(k_{5b}Cdc42D_m - k_{5a}Cdc42D_c) + D_c\Delta Cdc42D_c$$

$$\frac{\partial}{\partial t}BemGEF_c = \eta(k_{1b}BemGEF_m - k_{1a}BemGEF_c - k_7BemGEFc \cdot Cdc42T) + D_c\Delta BemGEF_c$$

| Description | Parameter | Value | Unit | Reference |
|---|---|---|---|---|
| $BemGEF_c$→ $BemGEF_m$ | $k_{1a}$ | 10 | $s^{-1}$ | *Goryachev and Pokhilko, 2008* |
| $BemGEF_m$→ $BemGEF_c$ | $k_{1b}$ | 10 | $s^{-1}$ | *Goryachev and Pokhilko, 2008* |
| $Cdc42D_m$ + $BemGEF_m$→ $Cdc42T_m$ + $BemGEF_m$ | $k_{2a}$ | 0.16 | $\mu M^{-1}\ s^{-1}$ | *Howell et al., 2009* |
| Cdc42T → $Cdc42D_m$ | $k_{2b}$ | 1.75 | $s^{-1}$ | *Wu et al., 2015* |

*Continued on next page*

*Continued*

| Description | Parameter | Value | Unit | Reference |
|---|---|---|---|---|
| Cdc42D$_m$ + BemGEF42 → Cdc42T + BemGEF42 | $k_3$ | 0.35 | μM$^{-1}$ s$^{-1}$ | *Howell et al., 2009* |
| BemGEF + Cdc42T → BemGEF42 | $k_{4a}$ | 10 | μM$^{-1}$ s$^{-1}$ | *Goryachev and Pokhilko, 2008* |
| BemGEF42 → BemGEF + Cdc42T | $k_{4b}$ | 10 | s$^{-1}$ | *Goryachev and Pokhilko, 2008* |
| Cdc42D$_c$→ Cdc42D$_m$ | $k_{5a}$ | 36 | s$^{-1}$ | *Kuo et al., 2014* |
| Cdc42D$_m$→ Cdc42D$_c$ | $k_{5b}$ | 0.65 | s$^{-1}$ | *Kuo et al., 2014* |
| BemGEF$_c$ + Cdc42T → BemGEF42 | $k_7$ | 10 | μM$^{-1}$ s$^{-1}$ | *Goryachev and Pokhilko, 2008* |
| Diffusion constant on the membrane | D$_m$ | 0.0025 | μm$^2$ s$^{-1}$ | *Goryachev and Pokhilko, 2008* |
| Diffusion constant in the cytoplasm | D$_c$ | 10 | μm$^2$ s$^{-1}$ | *Goryachev and Pokhilko, 2008* |
| Membrane to cytoplasm volume ratio | η | 0.01 | | *Goryachev and Pokhilko, 2008* |

In addition, the yeast polarity circuit contains a negative feedback loop due to multi-site phosphorylation of the GEF by the PAK, causing inactivation of the GEF (*Kuo et al., 2014*). Phosphorylation occurs when the PAK from one complex phosphorylates the GEF from another complex, which only happens when both complexes are bound to GTP-Cdc42. Dephosphorylation occurs only in the cytoplasm. The phosphorylated species, BemGEF*, can still exchange between cytoplasmic (BemGEF*$_c$) and membrane (BemGEF*$_m$) forms, and bind reversibly to Cdc42T (generating BemGEF*42). The addition of negative feedback leads to the differences highlighted in bold:

$$\frac{\partial}{\partial t}Cdc42T = (k_{2a}BemGEF_m + k_3BemGEF42) \cdot Cdc42D_m - (k_{2b} + k_{4a}\textbf{BemGEF}_{\textbf{mt}} + k_7\textbf{BemGEF}_{\textbf{ct}}) \cdot Cdc42T + k_{4b}\textbf{BemGEF42}_{\textbf{t}} + D_m\Delta Cdc42T$$

$$\frac{\partial}{\partial t}Cdc42D_m = k_{2b}Cdc42T - (k_{2a}BemGEF_m + k_3BemGEF42) \cdot Cdc42D_m - k_{5b}Cdc42D_m + k_{5a}Cdc42D_c + D_m\Delta Cdc42D_m$$

$$\frac{\partial}{\partial t}BemGEF42 = (k_{4a}BemGEF_m + k_7BemGEF_c) \cdot Cdc42T - (k_{4b} + \textbf{k}_{\textbf{8}}\textbf{BemGEF42}) \cdot BemGEF42 + D_m\Delta BemGEF42$$

$$\frac{\partial}{\partial t}BemGEF_m = k_{1a}BemGEF_c - k_{1b}BemGEF_m + k_{4b}BemGEF42 - k_{4a}BemGEF_m \cdot Cdc42T + D_m\Delta BemGEF_m$$

$$\frac{\partial}{\partial t}Cdc42D_c = \eta(k_{5b}Cdc42D_m - k_{5a}Cdc42D_c) + D_c\Delta Cdc42D_c$$

$$\frac{\partial}{\partial t}BemGEF_c = \eta(k_{1b}BemGEF_m - k_{1a}BemGEF_c - k_7BemGEF_c \cdot Cdc42T) + \textbf{k}_{\textbf{9}}\textbf{BemGEF}^{*}_{\textbf{c}} + D_c\Delta BemGEF_c$$

$$\frac{\partial}{\partial t}\textbf{BemGEF}^{*}_{\textbf{m}} = \textbf{k}_{\textbf{1a}}\textbf{BemGEF}^{*}_{\textbf{c}} - \textbf{k}_{\textbf{1b}}\textbf{BemGEF}^{*}_{\textbf{m}} + \textbf{k}_{\textbf{4b}}\textbf{BemGEF}^{*}\textbf{42} - \textbf{k}_{\textbf{4a}}\textbf{BemGEF}^{*}_{\textbf{m}} \cdot \textbf{Cdc42T} + \textbf{D}_{\textbf{m}}\Delta\textbf{BemGEF}^{*}_{\textbf{m}}$$

$$\frac{\partial}{\partial t}\textbf{BemGEF}^{*}_{\textbf{c}} = \eta(\textbf{k}_{\textbf{1b}}\textbf{BemGEF}^{*}_{\textbf{m}} - \textbf{k}_{\textbf{1a}}\textbf{BemGEF}^{*}_{\textbf{c}} - \textbf{k}_{\textbf{7}}\textbf{BemGEF}^{*}_{\textbf{c}} \cdot \textbf{Cdc42T}) - \textbf{k}_{\textbf{9}}\textbf{BemGEF}^{*}_{\textbf{c}} + \textbf{D}_{\textbf{c}}\Delta\textbf{BemGEF}^{*}_{\textbf{c}}$$

$$\frac{\partial}{\partial t}\textbf{BemGEF}^{*}\textbf{42} = (\textbf{k}_{\textbf{4a}}\textbf{BemGEF}^{*}_{\textbf{m}} + \textbf{k}_{\textbf{7}}\textbf{BemGEF}^{*}_{\textbf{c}}) \cdot \textbf{Cdc42T} - \textbf{k}_{\textbf{4b}} \cdot \textbf{BemGEF}^{*}\textbf{42} + \textbf{k}_{\textbf{8}}\textbf{BemGEF42}_{\textbf{t}} \cdot \textbf{BemGEF42} + \textbf{D}_{\textbf{m}}\Delta\textbf{BemGEF}^{*}\textbf{42}$$

Note that the subscript 't' is used to denote the sum of phosphorylated and unphosphorylated species, which can both undergo reversible binding to either membranes or GTP-Cdc42.

$$BemGEF_{mt} = BemGEF_m + BemGEF^{*}_m$$

$$BemGEF_{ct} = BemGEF_c + BemGEF_c^*$$

$$BemGEF42_t = BemGEF42 + BemGEF^*42$$

Also, although only unphosphorylated species can act as GEFs, both phosphorylated and unphosphorylated complexes can act as PAKs to phosphorylate other GEFs. Multi-site phosphorylation and dephosphorylation are assumed to occur in an ultrasensitive manner.

$$k_8 = k_{8max} \frac{BemGEF_{mt}^{k_{8n}}}{k_{8h}^{k_{8n}} + BemGEF_{mt}^{k_{8n}}}$$

$$k_9 = k_{9max} \frac{BemGEF_{mt}^{*k_{9n}}}{k_{9h}^{k_{9n}} + BemGEF_c^{*k_{9n}}}$$

| Description | Parameter | Value | Unit | Reference |
|---|---|---|---|---|
| $BemGEF_c \rightarrow BemGEF_m$ <br> $BemGEF_c^* \rightarrow BemGEF_m^*$ | $k_{1a}$ | 10 | $s^{-1}$ | *Goryachev and Pokhilko, 2008* |
| $BemGEF_m \rightarrow BemGEF_c$ <br> $BemGEF_m^* \rightarrow BemGEF_c^*$ | $k_{1b}$ | 10 | $s^{-1}$ | *Goryachev and Pokhilko, 2008* |
| $Cdc42D_m + BemGEF_m \rightarrow$ <br> $Cdc42T_m + BemGEF_m$ | $k_{2a}$ | 0.16 | $\mu M^{-1} s^{-1}$ | *Howell et al., 2009* |
| $Cdc42T \rightarrow Cdc42D_m$ | $k_{2b}$ | 0.35 | $s^{-1}$ | *Wu et al., 2015* |
| $Cdc42D_m + BemGEF42 \rightarrow$ <br> $Cdc42T + BemGEF42$ | $k_3$ | 0.35 | $\mu M^{-1} s^{-1}$ | *Howell et al., 2009* |
| $BemGEF_m + Cdc42 \rightarrow BemGEF42$ <br> $BemGEF_m^* + Cdc42 \rightarrow BemGEF42$ | $k_{4a}$ | 10 | $\mu M^{-1} s^{-1}$ | *Goryachev and Pokhilko, 2008* |
| $BemGEF42 \rightarrow BemGEF_m^* + Cdc42T$ <br> $BemGEF^*42 \rightarrow BemGEF_m^* + Cdc42T$ | $k_{4b}$ | 10 | $s^{-1}$ | *Goryachev and Pokhilko, 2008* |
| $Cdc42D_c \rightarrow Cdc42D_m$ | $k_{5a}$ | 36 | $s^{-1}$ | *Kuo et al., 2014* |
| $Cdc42D_m \rightarrow Cdc42D_c$ | $k_{5b}$ | 0.65 | $s^{-1}$ | *Kuo et al., 2014* |
| $BemGEF_c + Cdc42T \rightarrow BemGEF42$ <br> $BemGEF_c^* + Cdc42T \rightarrow BemGEF^*42$ | $k_7$ | 10 | $\mu M^{-1} s^{-1}$ | *Goryachev and Pokhilko, 2008* |
| $BemGEF42 + BemGEF42 \rightarrow$ <br> $BemGEF42 + BemGEF^*42$ <br> $BemGEF42 + BemGEF^*42 \rightarrow$ <br> $BemGEF^*42 + BemGEF^*42$ | $k_8$ | $k_{8max} = 0.0063$ <br> $k_{8n} = 6$ <br> $k_{8h} = 10$ | $\mu M^{-1} s^{-1}$ | *Kuo et al., 2014* |
| $BemGEF_c^* \rightarrow BemGEF_c$ | $k_9$ | $k_{9max} = 0.0044$ <br> $k_{8n} = 6$ <br> $k_{8h} = 0.003$ | $s^{-1}$ | *Kuo et al., 2014* |
| Diffusion constant on the membrane | $D_m$ | 0.0025 | $\mu m^2 s^{-1}$ | *Kuo et al., 2014* |
| Diffusion constant in the cytoplasm | $D_c$ | 10 | $\mu m^2 s^{-1}$ | *Goryachev and Pokhilko, 2008* |
| Membrane to cytoplasm volume ratio | $\eta$ | 0.01 | | *Goryachev and Pokhilko, 2008* |

## Numerical simulations

Simulations of the MCAS models were done on MATLAB. Simulations of conceptual models were done on one-dimensional domains with spatial resolution of 500 grid points. Finite differences were used with the linear diffusion being treated implicitly and the nonlinear reaction term explicitly in the time stepping. Mechanistic models were simulated on two-dimensional domains with 100 × 100 grid points. All simulations proceeded with adaptive time stepping according to relative error in the reaction term. Initial conditions for simulations of two unequal peaks were standardized by simulating two insulated subsystems containing 60% and 40% of the total mass. After they reached steady

state, the two subsystems were allowed to communicate by diffusion. The MATLAB code used for simulations is provided in Source Code Files.

## Acknowledgements

We thank Tim Elston, Erwin Frey, Nick Buchler, Stefano Di Talia, Amy Gladfelter, Masayuki Onishi, and members of the Lew lab for comments on the manuscript. Thanks to Trevin Zyla for help with yeast strain construction. This work was funded by NIH/NIGMS grant R35GM122488 to DJL.

## Additional information

### Funding

| Funder | Grant reference number | Author |
|---|---|---|
| National Institutes of Health | MIRA R35GM122488 | Daniel J Lew |

The funders had no role in study design, data collection and interpretation, or the decision to submit the work for publication.

### Author contributions

Jian-geng Chiou, Conceptualization, Resources, Data curation, Software, Formal analysis, Investigation, Visualization, Methodology, Writing - original draft, Project administration, Writing - review and editing; Kyle D Moran, Data curation, Formal analysis, Investigation, Validation, Visualization, Writing - review and editing; Daniel J Lew, Conceptualization, Resources, Supervision, Funding acquisition, Validation, Writing - original draft, Project administration, Writing - review and editing

### Author ORCIDs

Jian-geng Chiou (iD) https://orcid.org/0000-0003-3246-5841
Daniel J Lew (iD) https://orcid.org/0000-0001-7482-3585

### Decision letter and Author response

Decision letter https://doi.org/10.7554/eLife.58768.sa1
Author response https://doi.org/10.7554/eLife.58768.sa2

## Additional files

### Supplementary files

• Source code 1. Mechanistic negative feedback model. The source code files used to simulate the 2D mechanistic negative feedback model based on GEF phosphorylation (*Figure 6Aiii*, B) and 2D mechanistic positive feedback model (*Figure 1GHIJ*) if k8 is set to zero.

• Source code 2. Indirect substrate model. The source code files used to simulate 1D conceptual 'Indirect substrate model' in *Figure 5*.

• Transparent reporting form

### Data availability

All data generated or analyses during this study are included in the manuscript and supporting files.

The following datasets were generated:

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
