## [Decision Letter]

**Acceptance summary:**

Your paper using genetics and imaging (with an embedded computational framework) very nicely examines how the number of polarity sites are established, using yeast as a model. The dissection of competition mechanism and the novel equalization that you describe will fuel further work in this field both in yeast and in other organisms, especially those which contain multiple polarity sites in normal physiology.

**Decision letter after peer review:**

[Editors’ note: the authors submitted for reconsideration following the decision after peer review. What follows is the decision letter after the first round of review.]

Thank you for submitting your work entitled "How cells determine the number of polarity sites" for consideration by *eLife*. Your article has been reviewed by 3 peer reviewers, and the evaluation has been overseen by a Reviewing Editor and a Senior Editor. The reviewers have opted to remain anonymous.

Our decision has been reached after consultation between the reviewers. Based on these discussions and the individual reviews below, we regret to inform you that your work will not be considered further for publication in *eLife*.

The referees are enthusiastic about the topic and the question you are addressing, i.e. how polarity sites and their numbers are chosen. They also found a number of observations in the work interesting. However, all the three independent referees have raised a number of questions, both concerning the modeling as well as with the experiments, and also in citation of past work. In light of all these concerns, we are unable to publish this work in *eLife*. The referees comments are provided verbatim below.

*Reviewer #1:*

The manuscript entitled "How cells determine the number of polarity sites" by Chiou et al., compares theoretical models for polarity establishment in budding yeast, specifically their regulatory mechanisms, with experimental observations, in particular those that can lead to polarity cluster coexistence or equalization. After examining the implications of several models of increasing complexity the predictions of specific aspects of these models are explored, in particular alteration of cell size and polarity protein levels. While the findings presented are interesting, the overall message is not sufficiently clear and hence this obscures the significance of the work. Substantial rewriting is necessary together with refocusing to make the work more accessible to a broad audience.

1. The main findings of this work are not evident from the title, introduction and first paragraph of the Discussion. The Discussion talks about several classes of models, but does not indicate clearly which model best describes the observed behaviour. The title and abstract are somewhat general, overall appear descriptive and do not clearly indicate the findings. For example, in the Discussion section, page 25, it states, "Here we propose a novel mechanism for equalization that does not require negative feedback, but can account for the behavior of the more complex models that incorporate negative feedback," however it is not evident what this mechanism is.

2. Reference to and discussion of finding from several relevant and recent studies are lacking. For example, 3 studies, which use optogenetic systems to alter levels and/or clusters of active Cdc42 in fungi are not discussed, these findings are likely to be relevant to mechanism of polarity cluster control, see

-Witte et al., e*Life* 2017 e26722

-Lamas et al., Plos Biol 2020 18: e3000600

-Silva et al., Cell Rep 2019 28: 2231

Note that while the latter two are carried out in different yeast, the approaches involve temporal recruitment of active Cdc42 to the plasma membrane and hence are relevant to this work.

Furthermore the Discussion section (to some extent also the Introduction section, see also below), including the last subsection of "Implications for other systems" unfortunately does not put this study into the context of what is known about polarity determination outside of the budding yeast perspective, which limits its general interest. For example, I am surprised there is no mention or discussion of the recent work from the Goehring laboratory entitled "A Cell Size Threshold Limits Cell Polarity and Asymmetric Division Potential" (Hubatsch et al., Nat Phys 2019 15:1075) which would appear to be very relevant.

Similarly, the introduction seems to be overly focused on budding yeast with little indication of other systems and the more recent findings, including those from fission yeast (as opposed to cursory mention of an old *S. pombe* review) and other fungi, such as *Neurospora crassa*, among others. Furthermore, other reviews cited are quite old.

3. Throughout, the terminology 'protein content' is used which is imprecise. Are the authors referring to total protein amount or amount divided by volume, i.e. an effective concentration? This ambiguity is confusing, in particular in the Discussion section. It is unclear whether larger cells (page 20) have a higher amount or effective concentration of polarity proteins. It is surprising that this aspect of polarity would be cell size dependent given different sizes of haploid and diploid yeast cells.

4. Figures should be organised more clearly, with panels in left to right, top to bottom order (Figure 1-3), graph axes indicated (lacking in many panels including 1D, 2D, E, G, H), color schemes unclear (1G and 1H compared to 1I and 1J; green color used for different species), what is shown in different panels is not indicated (2D, 2H), in some scatter plots means and standard deviation should be indicated (3G, 5A, 5G), in several graphs error bars should be indicated (5B, 5E, 5I) and Figure 6 can be substantially simplified with at most 2 examples of each behaviour shown in A, together with an indication of what the lines refer to (the rest can go in the supporting figure). In addition, it is not clear in 6A if competed (note mix of past tense verbs and nouns for description, better to indicate 'competition, coexistence and equal) is actually different than equalized, in which the initial intensities are inversed, i.e. red higher than blue (A, i).

5. The analysis and interpretation of the cytoplasmic connection in Figure 3D-F and presented on page 14 (and mentioned again on pages 21 and 23) appears over-simplified. Firstly, diffusion of relatively small cytoplasmic GFP is unlikely to be directly relevant with respect to larger proteins and complexes, as well as those which can associate with membranes. Indeed this same septin mutant allele (cdc12-6) has been used to show that septins play an important role in cell cortex compartmentalization (Barral et al., 2000 Mol Cell 5:8410). Secondly, the photobleaching experiments were single photon, i.e. not limited to a small focal volume and hence a substantial region above and below the focal plane was bleached. It is unclear, in this situation, how differences between bud and mother cell geometry affects fluorescence recovery. As a result, attributing the outcomes to cell size appears to be an overly strong conclusion (page 14 and bottom of 15). Indeed the word cell geometry may be more appropriate than size. In the data presented in Figure 3H do the 2-budded cells form buds simultaneously or subsequently? This should be indicated and shown in a supporting figure.

*Reviewer #2:*

In this manuscript the authors study the concepts and cellular mechanisms that allow formation of multiple polarization sites. They focus on the *S. cerevisiae* model system and combine theoretical models with quantitative image data as validation of their predictions. They interpret formation of multiple polarization sites as a shift from competition to co-existence in a mass conserved reaction-diffusion system (MCAS). They conclude that a key feature in this scenario is the amount of available substrate in the system, determined by parameters such as cell size and expression levels.

My key conceptual criticism is that while the authors discuss several variants of reaction diffusion models in detail they completely ignore two fundamental aspects of cellular polarity systems: the cycling of the GTPase and in particular the link between activity cycling and physical cycling via GDI, and the role of vesicular transport in recycling of membrane-bound proteins and in supporting the polarization process. This is particularly striking as the role of these parameters has been studied in detail regarding their effects on formation of multiple polarization sites. As one key conclusion from previous work was that yeast cell were not able to form multiple polarization sites in the absence of actin I would have expected a convincing argument to ignore this point in the current study. Previous data also showed that the levels of active Cdc42 (by overexpressing Cdc24 or using a slow cycling mutant) directly increased the number of polarization sites formed through its limiting effect on GDI-based recycling. By ignoring those fundamental aspects of the polarity system the authors made it very hard for me to accept or follow the arguments provided in this study.

1. All citations for interactions between Cdc42, PAK Bem1 and Cdc24 are very selective – several of the postulated interactions are far from being established and open questions should be clearly stated. Lacks all referrals to pertinent studies by Li, McCusker and Wedlich-Söldner labs.

2. Page 6 third paragraph: in scenarios of two linked substrates isn't it obvious that the one with lower abundance will be limiting? Not really fitting to a Results section in my mind.

3. Page 10, third paragraph: Freisigner et al. is cited for the lack of effect of Cdc42 OE, but all the results in this study showing that OE of Cdc24 or increasing the activity of Cdc42 (deletion of Bem2 or fast cycling mutant of Cdc42) leads to formation of multiple polarization sites are ignored – those would be far more relevant here.

4. The image series in Figure 3A does not show two buds growing simultaneously from one cell segment – does this ever happen or are the buds only formed from distant segments? If diffusion is indeed not limiting shouldn't the distance between two forming buds be random – hence also occur within a single segment?

5. In general I have fundamental issues with the chosen method of generating larger cells – the cdc12 defective cells have not been sufficiently characterized and have man additional parameter changes beyond the simple increase in volume. Effects on cell cycle, attachment of formins (Bnr1 is recruited through septins), PM organization (what happens to eisosomes, lipid composition etc.) and many more. The cell is simply too complex to use such crude methods to validate mechanistic models. I didn't understand why they did not simply go with cell cycle arrest and larger cells. Even with the reported cytosolic dilution they could perform tests where they relate their conditions to the corresponding controls.

6. They should definitely show how actin is distributed in the cell chains and perform the basic tests of polarization in LatA treated cells. While Bem1 or Cdc24 might not be limited by diffusion – actin nucleators, vesicles or actin filaments will likely be. Of course bud formation will be stopped but polarized patches should still be able to form. The effect of 37{degree sign}C shift is also of particular relevance in this context as it has been linked to actin disruption in previous studies.

7. Please provide the bud-number analysis for the correct test strain with Drsr1 and expressed polarity marker – is this equivalent to the numbers in 3B? Please provide images for bud scars to exclude effects of those (even in rsr1D) on polarization.

8. Formation of Bem1-clusters will likely depend on local lipid composition – images often not good enough to distinguish between local micro clusters and polarized accumulation but the clusters often seem to be at the base of the bud and not at the bud tip – please clarify.

9. Panels in Figure 4 are too small – nearly impossible to follow. Blue patch in lower series of 4B seems to move from right to left – possible issues with projection or deconvolution? Why is patch not at bud tip? Same in lower series of 4C: patch seems to be at base of bud – very confusing.

10. Figure 5I: those three proteins cannot be thrown together – OE of Cdc42 and Cdc24 should result in very different outcomes (change in substrate or activator) – provide effects for each protein separately and show actual images as well as quantification of protein levels (western or GFP fluorescence). Again, this has been done in similar way in Freisinger et al. and showed link between Cdc42 GTPase cycle and number of polarity sites.

*Reviewer #3:*

This is a dense and quite technical manuscript pertaining to the mechanism of cell polarization in budding yeast which relies on a Rho family GTPase, Cdc42, that is known to exhibit positive feedback and in wild-type cells. The focus of this work is to identify whether these same biochemical circuit can also generate two foci that do not undergo competition as is the norm in this pathway, and if so, to identify such conditions and to provide a conceptual framework for the absence of competition.

In its present state, I do not find that this manuscript provides compelling evidence to support the underlying conceptual argument, namely that patch saturation can allow to foci to co-exist. Furthermore, the manuscript is challenging to follow, the figures are sparsely annotated so they are not self-explanatory and the text, while readable, is to lengthy (almost 10K words) not well organized, diluting the authors message.

Comments related to modeling:

1. It is not clear why the authors discuss the minimalistic model in this paper. It is discussed in their earlier work, which is probably essential for many readers to understand this manuscript (or superfluous for those fully conversant in these models). I understand that it is for simplicity, but it is a distraction that does not apply to the in vivo situation. In addition, in figure 2C, the indirect substrate is a theoretical construct that alters the behavior of the model but lacks a mechanistic counterpart. Along these lines, there is some overlap with the Chiou, 2018 PLOS comp bio paper.

2. Does the authors model predict that equivalent cells would form variable numbers of patches (eg Figure 5B)? And that these patches would resolve with different outcomes (eg Figure 5E)? If not, what are the implications of this mismatch?

3. There is no direct experimental evidence to suggest the existence of an "indirect substrate" with differential mobility.

Comments related to experimental data:

1. The coexistence of multiple sites is predicated on the concept of saturation which has been shown to exist in simplified computational models. Based on that model, the authors explore some of the predictions of this model, namely that cell size and protein levels will enhance co-existence of multiple sites. However, the authors do not directly document saturation, which is the key predicate of the model. The predictions that are tested to support this model may not be unique to the proposed model.

2. The use of GFP and its diffusion as a model for all the relevant species in the model is not well substantiated. The relevant proteins are much larger, in variably stable protein-protein complexes, and associate with cortical factors, lipids, proteins etc. This is a critical point, because slower diffusion of key components could prevent patches positioned at a distance from effectively competing with one another. The diffusion rates of the relevant proteins would need to be measured directly, when expressed at their normal levels. Perhaps the authors could FRAP Bem1 at one patch and measure the rate at which the other patch dims.

3. Related to the previous point, in Figure 6 in the author's 2018 modeling paper, they suggest that when saturation is operative, an increase in cell size rapidly limits competition between patches (black curve). The change in behavior with increasing cell size is far less striking in vivo than would be predicted.

4. The authors provide evidence that the situation in vivo is far more complex that their model would predict. Pools of "cytokinetic Bem1" are documented and the position of bud sites is not random ("the locations at which patches formed were non-random, with a preference for bud tips and mother cell locations (Figure 5C)"). These phenomena indicate that distinct mechanisms may be operable in vivo raising doubts that the in silico version can be directly applied in vivo.

5. In the model, the authors appear to assume that all Bem1 is bound to Cdc24 and Cla4? This is not well supported by the evidence in the literature (page 14 refs).

6. The authors "utilized cytokinesis-defective yeast mutants to obtain large connected cells that continue cycling and presumably retain a normal overall protein composition." the underlying presumption is important for the analysis of the data, yet it is assumed, not tested.

7. For some of the cells in the second row of figure 6A, the linear "interpolation" does not match the data well. What is the basis for this "interpolation"?

It would be useful to indicate the distance between the patches for the 20 cells in Figure 6.

8. Inhibition of actin polymerization could be used to inhibit budding and cytokinesis and might allow the authors to obtain cells that continue to cycle but retain a simple geometry.

[Editors' note: further revisions were suggested prior to acceptance, as described below.]

Thank you for resubmitting your work entitled "How cells determine the number of polarity sites" for further consideration by *eLife*. Your revised article has been evaluated by Naama Barkai (Senior Editor) and a Reviewing Editor (Mohan Balasubramanian).

The manuscript has been improved but there are some remaining issues that need to be addressed, as outlined below. Please note that one experiment has been suggested (point 5 below). If you have the data readily available, please do add it.

The rest of the points raised can be addressed through rewriting.

1. The text related to the portion regarding saturation requires further editing:

Line 76 "However, recent studies have shown that the growth rate of a peak "saturates" as the activator in the peak exceeds a threshold." Given that saturation is not shown empirically, this sentence should refer to "modeling studies".

This paragraph also refers to the possibility of equalization and explicitly states on line 83 "it has not yet been determined whether equalization occurs in cells". Given that saturation has also not been empirically demonstrated in cells, this too should be explicitly stated.

2. The authors observe that some cells also exhibit equalization, which the authors indicate is not possible in the "mechanistic model" as written. They further show that this becomes possible if an "indirect substrate" is incorporated into the model. This is an interesting finding even though the "indirect substrate" remains hypothetical/speculative. However, it does reveal that the mechanistic model based on MCAS does not provide a fully accurate description of the in vivo situation. Given that the reader was previously led to believe that the mechanistic MCAS model was supported by the available data, this apparent shortcoming is a bit jarring. The authors state (line 333) "Our conclusions from analysis of the simple indirect substrate model can explain the outcomes from all of the models discussed above and in previous studies (Figure 5-Figure supp. 1)." This sentence suggests that models with an indirect substrate would also predict that the increase in protein concentration would also lead to multibudding in this regime and figure 5 supports this interpretation. Nevertheless, it would help the reader to even more explicitly state that indirect substrate models is fully consistent with the findings up to that point in the manuscript.

More broadly, this manuscript begins as a Figure 1 Theory paper in the Phillips vernacular (PMID 26584768), but transitions to a "Figure 7 Theory paper". It would behove the reader to more clearly foreshadow the revision to the model in a subsequent figure.

3. Moreover, the description of the indirect substrate is unclear: line 280ff "The GAP and the inhibited GEFi are neither substrates nor activators, and appear to play different roles in the polarity circuit. However, we noticed that they both provide a source of substrate: the GAP converts local GTP-Cdc42 into the substrate GDP-Cdc42, while the inhibited GEFi turns into the substrate GEF upon dephosphorylation. Thus, in both cases a new species produced by the activator is highly mobile and generates a substrate in the cytoplasm."

In particular, the phrase "the inhibited GEFi turns into the substrate GEF upon dephosphorylation" is unclear. While it is evident that the inhibited GEF turns into a GEF upon dephosphorylation, this active GEF is not a substrate and thus does not fit the descriptor of being "highly mobile and generates a substrate in the cytoplasm." particularly as the GEF is unlikely to be active until it reaches the membrane. The confusion appears to arise from a conflation of the minimalistic model with a semi-mechanistic one. Whereas the activator and substrate are interconvertible in the minimalistic model, the same is not true in the semi-mechanistic one. This requires clarification.

4. Along these lines, it would be appropriate for the authors to discuss in this context the paper from Rodriguez et al., PMID 28781174 which demonstrates that the anterior PAR complex proteins PAR-6 and aPKC are induced to dissociate from PAR-3 by Cdc42 binding, which is analogous to this principle.

5. This study also raises the question as to whether the presence of multiple initial polarity sites described in Howell, 2012 and subsequent papers from the lab results from the hydroxyurea treatment which, by creating a cell cycle delay, would be expected to increase the amount of polarity proteins that exist in cells, thereby facilitating such behavior as shown here. It would be quite interesting to determine whether the strain overexpressing Bem1, Cdc42, and Cdc24 generate more nascent sites in otherwise unperturbed G1 cells as compared to their WT counterparts.

---

## [Author Response]

[Editors’ note: The authors appealed the original decision. What follows is the authors’ response to the first round of review.]

Reviewer #1:The manuscript entitled "How cells determine the number of polarity sites" by Chiou et al., compares theoretical models for polarity establishment in budding yeast, specifically their regulatory mechanisms, with experimental observations, in particular those that can lead to polarity cluster coexistence or equalization. After examining the implications of several models of increasing complexity the predictions of specific aspects of these models are explored, in particular alteration of cell size and polarity protein levels. While the findings presented are interesting, the overall message is not sufficiently clear and hence this obscures the significance of the work. Substantial rewriting is necessary together with refocusing to make the work more accessible to a broad audience.

We apologize for not making our main findings clearer, and hope the revised version makes the overall messages clearer and more accessible.

1. The main findings of this work are not evident from the title, introduction and first paragraph of the Discussion. The Discussion talks about several classes of models, but does not indicate clearly which model best describes the observed behaviour. The title and abstract are somewhat general, overall appear descriptive and do not clearly indicate the findings. For example, in the Discussion section, page 25, it states, "Here we propose a novel mechanism for equalization that does not require negative feedback, but can account for the behavior of the more complex models that incorporate negative feedback," however it is not evident what this mechanism is.

To address these concerns, we have (i) reworked the Abstract to highlight the two main advances: testing MCAS model predictions in yeast, and elucidating a novel equalization mechanism in multi-component MCAS models. (ii) In the last paragraph of the Introduction, and the first paragraph of the Discussion, we summarize our main findings and indicate that our findings favor coexistence as the dominant reason for two-budded cells. (iii) We have explored model behaviors over a broad range of key parameters (see new bifurcation analysis in Figure 5), allowing a more accurate and insightful analysis of the basis for equalization. In the third section of the Discussion, which has been extensively re-written, we now provide a specific description of the key features of the indirect substrate that enable equalization. We hope that the equalization mechanism is now clear.

2. Reference to and discussion of finding from several relevant and recent studies are lacking. For example, 3 studies, which use optogenetic systems to alter levels and/or clusters of active Cdc42 in fungi are not discussed, these findings are likely to be relevant to mechanism of polarity cluster control, see-Witte et al., eLife 2017 e26722-Lamas et al., Plos Biol 2020 18: e3000600-Silva et al., Cell Rep 2019 28: 2231Note that while the latter two are carried out in different yeast, the approaches involve temporal recruitment of active Cdc42 to the plasma membrane and hence are relevant to this work.

These studies are indeed relevant, and they are now cited in the penultimate paragraph of the Introduction as well as the Results.

Furthermore the Discussion section (to some extent also the Introduction section, see also below), including the last subsection of "Implications for other systems" unfortunately does not put this study into the context of what is known about polarity determination outside of the budding yeast perspective, which limits its general interest. For example, I am surprised there is no mention or discussion of the recent work from the Goehring laboratory entitled "A Cell Size Threshold Limits Cell Polarity and Asymmetric Division Potential" (Hubatsch et al., Nat Phys 2019 15:1075) which would appear to be very relevant.

We have expanded the “Implications…” section of the Discussion to include more references including a discussion of Hubatsch et al.

Similarly, the introduction seems to be overly focused on budding yeast with little indication of other systems and the more recent findings, including those from fission yeast (as opposed to cursory mention of an old *S. pombe* review) and other fungi, such as Neurospora crassa, among others. Furthermore, other reviews cited are quite old.

The Introduction is focused on mathematical models of pattern formation: budding yeast are introduced in the last paragraph as one well-studied system in which experimental tests of model predictions can be carried out. Other fungal systems are mentioned in the last section of the Discussion.

3. Throughout, the terminology 'protein content' is used which is imprecise. Are the authors referring to total protein amount or amount divided by volume, i.e. an effective concentration? This ambiguity is confusing, in particular in the Discussion section. It is unclear whether larger cells (page 20) have a higher amount or effective concentration of polarity proteins. It is surprising that this aspect of polarity would be cell size dependent given different sizes of haploid and diploid yeast cells.

We apologize for not making this clear. We have revised the text to use unambiguous terms like “total amount of X in the system”. We now explicitly show using Western blots (Figure 4—Figure supp 1) that the concentration of polarity proteins does not change with cell size, and therefore that larger cells have a higher *total amount* of polarity proteins per cell.

4. Figures should be organised more clearly, with panels in left to right, top to bottom order (Figure 1-3), graph axes indicated (lacking in many panels including 1D, 2D, E, G, H), color schemes unclear (1G and 1H compared to 1I and 1J; green color used for different species), what is shown in different panels is not indicated (2D, 2H), in some scatter plots means and standard deviation should be indicated (3G, 5A, 5G), in several graphs error bars should be indicated (5B, 5E, 5I) and Figure 6 can be substantially simplified with at most 2 examples of each behaviour shown in A, together with an indication of what the lines refer to (the rest can go in the supporting figure). In addition, it is not clear in 6A if competed (note mix of past tense verbs and nouns for description, better to indicate 'competition, coexistence and equal) is actually different than equalized, in which the initial intensities are inversed, i.e. red higher than blue (A, i).

We have re-organized the Figures as suggested, added axis labels for all graphs, indicated the color scheme in the labels or Figure Legends, and added mean and standard deviations for scatter plots. We prefer to keep several examples in Figure 6; we have added more in a new Figure 6—Figure supp 1. The orange and blue in Figure 6 track intensities at different locations, so we can clearly distinguish whether initial inequalities between sites narrowed (implying equalization) or widened (implying competition). This is now explicitly stated in the Figure Legend.

5. The analysis and interpretation of the cytoplasmic connection in Figure 3D-F and presented on page 14 (and mentioned again on pages 21 and 23) appears over-simplified. Firstly, diffusion of relatively small cytoplasmic GFP is unlikely to be directly relevant with respect to larger proteins and complexes, as well as those which can associate with membranes. Indeed this same septin mutant allele (cdc12-6) has been used to show that septins play an important role in cell cortex compartmentalization (Barral et al., 2000 Mol Cell 5:8410). Secondly, the photobleaching experiments were single photon, i.e. not limited to a small focal volume and hence a substantial region above and below the focal plane was bleached. It is unclear, in this situation, how differences between bud and mother cell geometry affects fluorescence recovery. As a result, attributing the outcomes to cell size appears to be an overly strong conclusion (page 14 and bottom of 15). Indeed the word cell geometry may be more appropriate than size. In the data presented in Figure 3H do the 2-budded cells form buds simultaneously or subsequently? This should be indicated and shown in a supporting figure.

Our conclusion from the bleaching experiment is that diffusion across the neck is mildly impaired, to the same degree as would be predicted from the geometry (i.e. the narrowing at the neck). To make this clear, we now present the 3D simulation of the experiment first, and then the data (New Figure 2-Figure supp 2). Our goal with this experiment was not to extract the diffusion constant, but to determine whether some blockage occurred at the neck. The match between result and model expectation indicates that there are no additional unexpected barriers to diffusion across the neck. Barral et al., (2000) concluded that septins retard diffusion of cortical proteins and that *cdc12-*6 removed this barrier, which is fully consistent with our NOT finding a barrier in *cdc12-6* cells.

With respect to attributing the outcomes to cell size: Even the small effect of neck geometry on the diffusional connection between compartments has the potential to promote 2-budded outcomes. In distinguishing whether retarded diffusion through the neck or cell size was the dominant factor in promoting 2-budded outcomes, we relied on the comparison between haploid (smaller neck, smaller cell) and diploid cells (larger neck, larger cell). Smaller necks (haploid) would retard diffusion more effectively, yet haploids produced fewer 2-budded outcomes, leading us to favor size as the main contributor. To clarify these points we moved the bleaching experiment to a supplementary Figure and highlighted the haploid/diploid comparison in the text.

We have added Video 1 to show that the buds of 2-budded cells form simultaneously (within the resolution of our time-lapse). This is explicitly stated in the text.

Reviewer #2:In this manuscript the authors study the concepts and cellular mechanisms that allow formation of multiple polarization sites. They focus on the *S. cerevisiae* model system and combine theoretical models with quantitative image data as validation of their predictions. They interpret formation of multiple polarization sites as a shift from competition to co-existence in a mass conserved reaction-diffusion system (MCAS). They conclude that a key feature in this scenario is the amount of available substrate in the system, determined by parameters such as cell size and expression levels.My key conceptual criticism is that while the authors discuss several variants of reaction diffusion models in detail they completely ignore two fundamental aspects of cellular polarity systems: the cycling of the GTPase and in particular the link between activity cycling and physical cycling via GDI, and the role of vesicular transport in recycling of membrane-bound proteins and in supporting the polarization process. This is particularly striking as the role of these parameters has been studied in detail regarding their effects on formation of multiple polarization sites. As one key conclusion from previous work was that yeast cell were not able to form multiple polarization sites in the absence of actin I would have expected a convincing argument to ignore this point in the current study. Previous data also showed that the levels of active Cdc42 (by overexpressing Cdc24 or using a slow cycling mutant) directly increased the number of polarization sites formed through its limiting effect on GDI-based recycling. By ignoring those fundamental aspects of the polarity system the authors made it very hard for me to accept or follow the arguments provided in this study.

With regard to GTPase cycling and GDI: we agree that these are major key parameters—they are incorporated into ALL of the models we discussed and referenced in the papers now cited, including the optogenetic papers suggested by reviewer #1.

With regard to the role of vesicular transport in recycling Cdc42 to support polarization: we did not focus on historical differences in findings and interpretations between our lab and the labs of Rong Li and Roland Wedlich-Soldner, as we believe that they have been largely addressed by prior work (as discussed for instance in (Woods and Lew, 2017)). Some of those are relevant to the reviewer<milestone-start />’<milestone-end />s comments, and we discuss them briefly below.

While there is agreement on the presence of two-budded cells in rdi1 mutants lacking the GDI, the origin of the two-budded cells has been controversial. Our interpretation is that deleting the GDI slows, but does not stop, membrane-cytoplasm exchange of Cdc42 (Woods et al., 2016), that slowed exchange slows the competition between polarity sites (Wu et al., 2015), and that slowed competition allows polarity sites to persist past bud emergence, yielding >1 bud. Consistent with that view, artificially slowing membrane-cytoplasm exchange of Bem1 or Cdc24 also slowed competition and led to the appearance of cells with two or more buds (Wu et al., 2015).

Wedlich-Soldner and Li were the first to show that Cdc42 recycling was slowed in rdi1 mutants, but they assumed that in those mutants the Cdc42 was unable to detach from the membrane, and hence that all remaining recycling was due to actin-mediated vesicle traffic (Freisinger et al., 2013; Slaughter et al., 2009). Consistent with that view, they reported that cells were unable to polarize Cdc42 in the combined absence of Rdi1 and F-actin (Freisinger et al., 2013; Smith et al., 2013). Freisinger et al. concluded that two-budded cells arose from reliance on actin-mediated traffic in rdi1 mutants, which was reasonable given their findings and assumptions at the time.

However, our subsequent findings argue strongly that even in rdi1 mutants, F-actin is not required to polarize Cdc42 or Bem1 or Cdc24, that actin cables do not affect the timing or efficiency of polarization, and that Cdc42 can still exchange (albeit more slowly) between membrane and cytoplasm (Woods et al., 2016). These findings as well as several earlier findings led us to conclude that F-actin is not important for polarizing Cdc42 (Dyer et al., 2013; Howell et al., 2012; Howell et al., 2009; Irazoqui et al., 2003; Layton et al., 2011; Savage et al., 2012), reviewed in (Chiou et al., 2017; Johnson et al., 2011; Woods and Lew, 2017).

1. All citations for interactions between Cdc42, PAK Bem1 and Cdc24 are very selective – several of the postulated interactions are far from being established and open questions should be clearly stated. Lacks all referrals to pertinent studies by Li, McCusker and Wedlich-Söldner labs.

We have added citations for the direct interaction between Cdc42 and PAKs (Bose et al., 2001; Cvrckova et al., 1995; Lamson et al., 2002; Zhao et al., 1995), PAKs and Bem1 (Bose et al., 2001; Leeuw et al., 1995), and Bem1 and Cdc24 (Bose et al., 2001; Butty et al., 2002; Ito et al., 2001; Peterson et al., 1994)(Rapali et al., 2017) in the penultimate paragraph of the Introduction. We are not aware of any controversy or open questions regarding these interactions.

2. Page 6 third paragraph: in scenarios of two linked substrates isn't it obvious that the one with lower abundance will be limiting? Not really fitting to a Results section in my mind.

We have moved those panels to the new Figure 1—supp Figure 1. Our point in showing the results of those simulations was in part that a substrate can become limiting even if it is *not* the one with lower abundance (note the different axis labels for the concentrations of Cdc42 or Bem1-GEF). Which substrate is limiting depends on the reaction parameters as well as abundance.

3. Page 10, third paragraph: Freisigner et al. is cited for the lack of effect of Cdc42 OE, but all the results in this study showing that OE of Cdc24 or increasing the activity of Cdc42 (deletion of Bem2 or fast cycling mutant of Cdc42) leads to formation of multiple polarization sites are ignored – those would be far more relevant here.

Actually Freisinger did not see an effect of OE of Cdc24 unless they also deleted *BEM2* , and even then the effect was marginal, though it could be enhanced by Latrunculin treatment/washout (see their Figure 7E)(Freisinger et al., 2013). We have added new experiments that individually doubled the gene dosage for *CDC42*, *CDC24*, or *BEM1*, and found that this did not necessarily increase multibudded outcomes (new Figure 4-supp Figure 2). We now cite Freisinger as being consistent with that result. Doubling the dosage for all three genes was more effective, and we have added our reasoning that this is expected because adding a single gene may alter the stoichiometry and make a different species limiting.

4. The image series in Figure 3A does not show two buds growing simultaneously from one cell segment – does this ever happen or are the buds only formed from distant segments? If diffusion is indeed not limiting shouldn't the distance between two forming buds be random – hence also occur within a single segment?

This is a good point. The two-budded cells we document have all emerged from different cell lobes (and in non-random locations, as we discussed). We suspect that the effect of cell geometry on diffusional connectivity, although too small to explain the existence of two-budded cells, may influence the placement of the buds and account for this phenomenon. This is now mentioned in the second section of the Discussion.

5. In general I have fundamental issues with the chosen method of generating larger cells – the cdc12 defective cells have not been sufficiently characterized and have man additional parameter changes beyond the simple increase in volume. Effects on cell cycle, attachment of formins (Bnr1 is recruited through septins), PM organization (what happens to eisosomes, lipid composition etc.) and many more. The cell is simply too complex to use such crude methods to validate mechanistic models. I didn't understand why they did not simply go with cell cycle arrest and larger cells. Even with the reported cytosolic dilution they could perform tests where they relate their conditions to the corresponding controls.

We acknowledge that any method to generate larger cells might inadvertently introduce unanticipated parameter changes, although we do not understand why the reviewer should feel that *cdc12-6* mutants are uniquely suspect in that regard. We have experimentally tested our assumption that overall concentrations of Cdc42, Cdc24, and Bem1 are unchanged by making cells larger in this way (new Figure 4-supp Figure 1).

With regard to cell cycle: we focus on initial polarization in late G1, so we do not believe that the extended G2 should affect the outcome. We found similar multi-budding in iqg1 depletion cells that still have intact septins and do not affect G2 (Figure 2-supp Figure 1B).

6. They should definitely show how actin is distributed in the cell chains and perform the basic tests of polarization in LatA treated cells. While Bem1 or Cdc24 might not be limited by diffusion – actin nucleators, vesicles or actin filaments will likely be. Of course bud formation will be stopped but polarized patches should still be able to form. The effect of 37{degree sign}C shift is also of particular relevance in this context as it has been linked to actin disruption in previous studies.

We see no reason to doubt that actin is polarized towards the same sites as Cdc42 in *cdc12-6* cells. With regard to Latrunculin: we view this as a much more disruptive treatment than *cdc12-6*. We note that Lat inhibits all endocytosis (Ayscough et al., 1997), so that plasma membrane composition changes a lot as material is added with no recycling. Furthermore, Lat treatment induces the cell wall integrity stress response (Harrison et al., 2001), which itself can cause depolarization. In fission yeast, Sawin and colleagues showed that the effects of Lat on Cdc42 (which are far more severe in that organism) are all mediated by a stress response (Mutavchiev et al., 2016). Thus, we believe that interpretation of Lat treatment is far more complex that generally assumed.

7. Please provide the bud-number analysis for the correct test strain with Drsr1 and expressed polarity marker – is this equivalent to the numbers in 3B? Please provide images for bud scars to exclude effects of those (even in rsr1D) on polarization.

The bud-number data for *cdc12-6rsr1*D mutants expressing the Bem1 probe is now shown in Figure 3-supp Figure 1. Like *cdc12-6 RSR1* strains, these exhibit multipolar outcomes as they grow larger. As expected from previous work showing that *RSR1* retards competition (Wu et al., 2013), *cdc12-6rsr1*D strains display fewer multipolar outcomes than *cdc12-6 RSR1* strains.

We did not understand how bud scar images would be informative. We note that septins are known to be essential for axial budding (Flescher et al., 1993), so axial budding is disabled with or without Rsr1 in our strains.

8. Formation of Bem1-clusters will likely depend on local lipid composition – images often not good enough to distinguish between local micro clusters and polarized accumulation but the clusters often seem to be at the base of the bud and not at the bud tip – please clarify.

We fully agree that lipid composition would affect the parameters of reaction-diffusion models and hence could impact competition between polarity sites. Detecting nanoclusters requires super-resolution methods: our spinning-disc confocal microscopy can detect protein-rich polarity sites but does not distinguish whether or not they are composed of nanoclusters. To remove this ambiguity, we no longer use the word <milestone-start />“<milestone-end />cluster” to describe polarity sites.

With regard to bud base or bud tip: once a bud is growing, the polarity site is always at the bud tip. This is now explicitly stated in the new Figure 6 Legend.

9. Panels in Figure 4 are too small – nearly impossible to follow. Blue patch in lower series of 4B seems to move from right to left – possible issues with projection or deconvolution? Why is patch not at bud tip? Same in lower series of 4C: patch seems to be at base of bud – very confusing.

We apologize for the small size and confusing display: the patches are always at the bud tips, but projection from 3D to 2D can create a misleading impression when the bud does not grow in the focal plane. In the new Figure 3 and associated supporting Figures the images are larger and explanatory cartoons have been added.

10. Figure 5I: those three proteins cannot be thrown together – OE of Cdc42 and Cdc24 should result in very different outcomes (change in substrate or activator) – provide effects for each protein separately and show actual images as well as quantification of protein levels (western or GFP fluorescence). Again, this has been done in similar way in Freisinger et al. and showed link between Cdc42 GTPase cycle and number of polarity sites.

We now show Western blots to quantify the expression levels of the proteins (new Figure 4-supp Figure 1), and examine the effect of doubling the dose for each protein individually (new Figure 4-supp Figure 2). This did not affect the frequency of multipolar outcomes much for any of the proteins. Overexpressing one component of a multi-component polarity system might fail to yield multi-polar outcomes because a non-overexpressed component becomes limiting. As noted above, Freisinger also did not observe any effect of doubling CDC24 dose unless combined with bem2 deletion.

Reviewer #3:This is a dense and quite technical manuscript pertaining to the mechanism of cell polarization in budding yeast which relies on a Rho family GTPase, Cdc42, that is known to exhibit positive feedback and in wild-type cells. The focus of this work is to identify whether these same biochemical circuit can also generate two foci that do not undergo competition as is the norm in this pathway, and if so, to identify such conditions and to provide a conceptual framework for the absence of competition.In its present state, I do not find that this manuscript provides compelling evidence to support the underlying conceptual argument, namely that patch saturation can allow to foci to co-exist. Furthermore, the manuscript is challenging to follow, the figures are sparsely annotated so they are not self-explanatory and the text, while readable, is to lengthy (almost 10K words) not well organized, diluting the authors message.

We have addressed these issues as follows:

– To reduce the density and length, we moved some sections to supplemental Figures, and reorganized the text to streamline the flow and remove unnecessary material.

– We summarized our main arguments and novel conclusions in the Abstract, the last paragraph of the Introduction, and the first paragraph of the Discussion. The conceptual argument identified by the reviewer, that “patch saturation can allow two foci to co-exist”, was actually the take-home from our previous paper on minimalistic models (Chiou et al., 2018). Here, our main advances are to show experimentally that yeast cells exhibit multipolar outcomes in a manner consistent with the MCAS models, to elucidate another mechanism (equalization) that can yield multipolar outcomes, and to distinguish which of the different model-based phenomena (periodic polarity peaks, slow competition/co-existence, or equalization) account for the multipolar outcomes in this system.

– We have better annotated the Figures and added explanatory cartoons to make them easier to follow.

Comments related to modeling:1. It is not clear why the authors discuss the minimalistic model in this paper. It is discussed in their earlier work, which is probably essential for many readers to understand this manuscript (or superfluous for those fully conversant in these models). I understand that it is for simplicity, but it is a distraction that does not apply to the in vivo situation. In addition, in figure 2C, the indirect substrate is a theoretical construct that alters the behavior of the model but lacks a mechanistic counterpart. Along these lines, there is some overlap with the Chiou, 2018 PLOS comp bio paper.

We discuss the minimalistic model precisely because it is essential for many readers to understand this manuscript. A basic run-down seemed advisable so that readers could understand the novel aspects of this paper, and devoting 50% of Figure 1 to such introductory material seemed appropriate. Based on our findings we do believe that it is applicable to the in vivo situation.

With regard to the indirect substrate: there are many potential mechanistic counterparts. We have extensively revised the section on equalization and now emphasize that the key is a pathway that indirectly converts activator to substrate. The last paragraph of the Discussion section on equalization now explicitly presents various examples of known candidates that could act in this manner. Our analysis does not overlap with Chiou, 2018, which dealt exclusively with 2-component models that lacked an indirect pathway from activator to substrate.

2. Does the authors model predict that equivalent cells would form variable numbers of patches (eg Figure 5B)? And that these patches would resolve with different outcomes (eg Figure 5E)? If not, what are the implications of this mismatch?

All of the models considered here are deterministic, and the outcomes would depend on initial conditions. Noise (fluctuations) in the spatial distribution of starting protein concentrations would determine how many initial patches form, so seeing different numbers of initial patches is entirely consistent with expectation. If 2 initial patches form, the rate of competition depends on model parameters and the (noise-based) initial asymmetry between the patches. These points are now explicitly discussed in the Results.

3. There is no direct experimental evidence to suggest the existence of a "indirect substrate" with differential mobility.

We now discuss several known polarity components that are candidates that could act as indirect substrates (last paragraph of the Discussion section on equalization).

Comments related to experimental data:1. The coexistence of multiple sites is predicated on the concept of saturation which has been shown to exist in simplified computational models. Based on that model, the authors explore some of the predictions of this model, namely that cell size and protein levels will enhance co-existence of multiple sites. However, the authors do not directly document saturation, which is the key predicate of the model. The predictions that are tested to support this model may not be unique to the proposed model.

The reviewer is correct that while our data indicate co-existence (rather than equalization) is the predominant reason for multipolar outcomes in yeast, we have not been able to demonstrate saturation from the fluorescent intensity profiles of the patches or the cytoplasm. A major issue is that when there are multiple species (as opposed to just one as in the minimalistic models), then which species is limiting depends on parameters, and modeling explains why the concentration profiles don’t show clear saturation (Figure 1-supp Figure 1). The more definitive changes in cytoplasmic concentrations (Figure 1) are unfortunately too minuscule to detect experimentally.

2. The use of GFP and its diffusion as a model for all the relevant species in the model is not well substantiated. The relevant proteins are much larger, in variably stable protein-protein complexes, and associate with cortical factors, lipids, proteins etc. This is a critical point, because slower diffusion of key components could prevent patches positioned at a distance from effectively competing with one another. The diffusion rates of the relevant proteins would need to be measured directly, when expressed at their normal levels. Perhaps the authors could FRAP Bem1 at one patch and measure the rate at which the other patch dims.

We note that because the dynamics of Bem1 coming on and off a patch are dominated by the relatively slow unbinding reaction, bleaching a patch would not provide any information about the relatively fast process of diffusion. Because competition between patches in different compartments is a frequent outcome, we do not doubt that Bem1 (and other polarity proteins) can travel between compartments. We do agree that Bem1 diffusion would be slower than that of GFP, but to do the analogous experiment (bleaching/recovery of cytoplasmic spot, not patch) with Bem1-GFP would require significant overexpression in order to get a decent signal:noise. We now clarify that our main conclusion from this experiment is simply that as predicted, neck geometry provides a minor but detectable impediment to diffusional communication.

3. Related to the previous point, in Figure 6 in the author's 2018 modeling paper, they suggest that when saturation is operative, an increase in cell size rapidly limits competition between patches (black curve). The change in behavior with increasing cell size is far less striking in vivo than would be predicted.

Figure 6 in Chiou et al., 2018 describes the deterministic outcome of competition between two standardized initial peaks containing 60% and 40% of the polarized proteins. However, the asymmetry of initial polarity sites in vivo is stochastic, and can range from 50:50 to 80:20. Stochasticity can obscure the effect of saturation, as in a deterministic model 80:20 sites can compete under the same parameters where 60:40 sites coexist. Another issue is that simulations on 2D surfaces (Figure 8A in Chiou, 2018 and Figure 1I in the current paper) show a more gradual increase in competition time than simulations of the same model in 1D domains (Figure 6 in Chiou, 2018). This is primarily because a secondary competition mechanism in 2D is still in effect even when peaks saturate, as discussed in Chiou, 2018. Thus, the in vivo behavior we observed is consistent with the 2D modeling predictions when one takes stochasticity of starting conditions into account.

4. The authors provide evidence that the situation in vivo is far more complex that their model would predict. Pools of "cytokinetic Bem1" are documented and the position of bud sites is not random ("the locations at which patches formed were non-random, with a preference for bud tips and mother cell locations (Figure 5C)"). These phenomena indicate that distinct mechanisms may be operable in vivo raising doubts that the in silico version can be directly applied in vivo.

Agreed! This is always an issue when trying to extract fundamental principles underlying complex phenomena, but we would argue that it does not invalidate the effort, and that conclusions that explain a large part (though not all) of the behavior are valuable.

5. In the model, the authors appear to assume that all Bem1 is bound to Cdc24 and Cla4? This is not well supported by the evidence in the literature (page 14 refs).

Agreed. This choice has a historical origin: in his foundational work Goryachev (Goryachev and Pokhilko, 2008) tested the effects of modeling separate species for Bem1 and GEF that could associate and dissociate, and concluded that the overall model behavior didn<milestone-start />’<milestone-end />t change (explained in the supplement of that paper), and we have followed his lead in simplifying the complex. More recently, Frey and colleagues (Klunder et al., 2013) also investigated a model with separate Bem1 and Cdc24 species; when examined in comparable parameter regimes (Wu et al., 2015), that model also predicts the same qualitative behavior. However, it is possible that in some parameter regimes, introducing separate PAK, Bem1, and GEF species would lead to equalization instead of competition, as the separate species have potential to act as indirect substrates. This is now addressed in the Discussion where the potential of separate GEF, Bem1, and PAK to act as indirect substrates is mentioned. In the Methods section providing a detailed description of the mechanistic model, we now clarify the historical reason for the use of a single species as well as the potential for more complex models with multiple species to behave differently.

6. The authors "utilized cytokinesis-defective yeast mutants to obtain large connected cells that continue cycling and presumably retain a normal overall protein composition." the underlying presumption is important for the analysis of the data, yet it is assumed, not tested.

Agreed. We now provide Western blot data confirming this assumption for Bem1, Cdc42, and Cdc24 (Figure 4-supp Figure 1).

7. For some of the cells in the second row of figure 6A, the linear "interpolation" does not match the data well. What is the basis for this "interpolation"?It would be useful to indicate the distance between the patches for the 20 cells in Figure 6.

We did not mean to imply that the process is linear. We have removed the linear trend lines to avoid confusion. And, we now indicate the distance between patches as an inset in each panel. We also added more example cells and plotted the distances between patches for all examples, showing that the outcome appears uncorrelated with inter-patch distance (Figure 6—supp Figure 1).

8. Inhibition of actin polymerization could be used to inhibit budding and cytokinesis and might allow the authors to obtain cells that continue to cycle but retain a simple geometry.

Latrunculin treatment causes cell cycle arrest in G2 (McMillan et al., 1998), so the cells would not continue cycling. Moreover, as mentioned in our response to reviewer 2 point 6, Latrunculin inhibits all endocytosis (Ayscough et al., 1997), so that plasma membrane composition changes a lot as material is added with no recycling. Furthermore, Lat treatment induces the cell wall integrity stress response (Harrison et al., 2001), which itself can cause depolarization. In fission yeast, Sawin and colleagues showed that the effects of Lat on Cdc42 (which are far more severe in that organism) are all mediated by a stress response (Mutavchiev et al., 2016). Thus, we believe that interpretation of Lat treatment is far more complex than generally assumed.

[Editors' note: further revisions were suggested prior to acceptance, as described below.]

The manuscript has been improved but there are some remaining issues that need to be addressed, as outlined below. Please note that one experiment has been suggested (point 5 below). If you have the data readily available, please do add it.The rest of the points raised can be addressed through rewriting.1. The text related to the portion regarding saturation requires further editing:Line 76 "However, recent studies have shown that the growth rate of a peak "saturates" as the activator in the peak exceeds a threshold." Given that saturation is not shown empirically, this sentence should refer to "modeling studies".

Good point. Edited as suggested.

This paragraph also refers to the possibility of equalization and explicitly states on line 83 "it has not yet been determined whether equalization occurs in cells". Given that saturation has also not been empirically demonstrated in cells, this too should be explicitly stated.

Agreed. Now edited.

2. The authors observe that some cells also exhibit equalization, which the authors indicate is not possible in the "mechanistic model" as written. They further show that this becomes possible if an "indirect substrate" is incorporated into the model. This is an interesting finding even though the "indirect substrate" remains hypothetical/speculative. However, it does reveal that the mechanistic model based on MCAS does not provide a fully accurate description of the in vivo situation. Given that the reader was previously lead to believe that the mechanistic MCAS model was supported by the available data, this apparent shortcoming is a bit jarring. The authors state (line 333) "Our conclusions from analysis of the simple indirect substrate model can explain the outcomes from all of the models discussed above and in previous studies (Figure 5-Figure supp. 1)." This sentence suggests that models with an indirect substrate would also predict that the increase in protein concentration would also lead to multibudding in this regime and figure 5 supports this interpretation. Nevertheless, it would help the reader to even more explicitly state that indirect substrate models is fully consistent with the findings up to that point in the manuscript.More broadly, this manuscript begins as a Figure 1 Theory paper in the Phillips vernacular (PMID 26584768), but transitions to a "Figure 7 Theory paper". It would behoove the reader to more clearly foreshadow the revision to the model in a subsequent figure.

We were not entirely sure what the editor had in mind here. Is the idea that the equalization instances in Figure 6D negate the mechanistic model in Figure 1F, necessitating an indirect substrate? That may be true, but the evidence for equalization is quite weak (only 3 cells, and only one timepoint in each instance distinguishes equalization from coexistence). We have changed the labeling in Figure 6D and added a sentence to the text (p. 9, bottom) to make this clear.

3. Moreover, the description of the indirect substrate is unclear: line 280ff "The GAP and the inhibited GEFi are neither substrates nor activators, and appear to play different roles in the polarity circuit. However, we noticed that they both provide a source of substrate: the GAP converts local GTP-Cdc42 into the substrate GDP-Cdc42, while the inhibited GEFi turns into the substrate GEF upon dephosphorylation. Thus, in both cases a new species produced by the activator is highly mobile and generates a substrate in the cytoplasm."In particular, the phrase "the inhibited GEFi turns into the substrate GEF upon dephosphorylation" is unclear. While it is evident that the inhibited GEF turns into a GEF upon dephosphorylation, this active GEF is not a substrate and thus does not fit the descriptor of being "highly mobile and generates a substrate in the cytoplasm." particularly as the GEF is unlikely to be active until it reaches the membrane. The confusion appears to arise from a conflation of the minimalistic model with a semi-mechanistic one. Whereas the activator and substrate are interconvertible in the minimalistic model, the same is not true in the semi-mechanistic one. This requires clarification.

Thank you for pointing out the lack of clarity in our discussion. In our view, the key features of an “activator” are that (i) the species has low mobility; and (ii) it can promote accumulation of more activator via positive feedback. Similarly, the key features of a “substrate” are that (i) the species has high mobility; and (ii) it can be converted into an activator. Using these criteria, the mechanistic model of Figure 1F has *two* species that could be considered as activators and *two* species that can be considered as corresponding substrates:

Activators: GTP-Cdc42 and Bem1-GEF-Cdc42. Both have low mobility and promote their own accumulation by positive feedback. There is no reason to distinguish only one of these as the “activator”.Substrates: GDP-Cdc42 and Bem1-GEF in the cytoplasm. Both have high mobility and can be converted into activators (GDP-Cdc42 is converted to the activator GTP-Cdc42, and Bem1-GEF is converted into the activator Bem1-GEF-Cdc42 when it binds GTP-Cdc42).

Accordingly, we believe it is correct to refer to the dephosphorylated GEF (cytoplasmic Bem1-GEF) as a substrate in Figure 5. We have edited the Results text (p. 4, top) to make this clearer. We have also edited the cartoon in Figure 5Aiii and corresponding text (p. 7) and Figure legend to enhance clarity.

4. Along these lines, it would be appropriate for the authors to discuss in this context the paper from Rodriguez et al. PMID 28781174 which demonstrates that the anterior PAR complex proteins PAR-6 and aPKC are induced to dissociate from PAR-3 by Cdc42 binding, which is analogous to this principle.

This is an interesting paper, whose model includes one species analogous to an activator (Cdc42-PAR6-aPKC at the membrane) and another analogous to a substrate (cytoplasmic PAR6-aPKC) as well as an intermediate species (PAR3-PAR6-aPKC) that is neither an activator nor a substrate. However, that species does not appear to be analogous to an indirect substrate. Note that the indirect substrate in our models is a species generated by an activator that can then create a substrate, whereas PAR3-PAR6-aPKC is generated from a substrate (PAR6-aPKC) and can then generate an activator (Cdc42-PAR6-aPKC). Moreover, the indirect substrate species we considered must have higher mobility than the activator (Figure 5), whereas the PAR3-PAR6-aPKC species has lower mobility than the activator. We have not explored models with these characteristics. As the key role of the PAR3-PAR6-aPKC species is to couple the reactant distribution to actomyosin flows, which are absent from all of the models we discuss, it seemed that comparison of that model with the ones we discuss is perhaps best left to a review.

5. This study also raises the question as to whether the presence of multiple initial polarity sites described in Howell, 2012 and subsequent papers from the lab results from the hydroxyurea treatment which, by creating a cell cycle delay, would be expected to increase the amount of polarity proteins that exist in cells, thereby facilitating such behavior as shown here. It would be quite interesting to determine whether the strain overexpressing Bem1, Cdc42, and Cdc24 generate more nascent sites in otherwise unperturbed G1 cells as compared to their WT counterparts.

This is an interesting question but we have not imaged that strain. We note that we have also seen multiple foci competing in mating cells (not treated with hydroxyurea), reported in a new paper: https://www.molbiolcell.org/doi/pdf/10.1091/mbc.E20-12-0757.